# Rapid evolution of bacterial mutualism in the plant rhizosphere

Erqin Li[1,2,3], Ronnie de Jonge [1,4,5✉], Chen Liu[1], Henan Jiang[1], Ville-Petri Friman [6✉], Corné M. J. Pieterse [1], Peter A. H. M. Bakker[1] & Alexandre Jousset [7✉]

While beneficial plant-microbe interactions are common in nature, direct evidence for the evolution of bacterial mutualism is scarce. Here we use experimental evolution to causally show that initially plant-antagonistic *Pseudomonas protegens* bacteria evolve into mutualists in the rhizosphere of *Arabidopsis thaliana* within six plant growth cycles (6 months). This evolutionary transition is accompanied with increased mutualist fitness via two mechanisms: (i) improved competitiveness for root exudates and (ii) enhanced tolerance to the plant-secreted antimicrobial scopoletin whose production is regulated by transcription factor *MYB72*. Crucially, these mutualistic adaptations are coupled with reduced phytotoxicity, enhanced transcription of *MYB72* in roots, and a positive effect on plant growth. Genetically, mutualism is associated with diverse mutations in the GacS/GacA two-component regulator system, which confers high fitness benefits only in the presence of plants. Together, our results show that rhizosphere bacteria can rapidly evolve along the parasitism-mutualism continuum at an agriculturally relevant evolutionary timescale.

[1] Utrecht University, Department of Biology, Plant-Microbe Interactions, Utrecht, The Netherlands. [2] Freie Universität Berlin, Institut für Biologie, Berlin, Germany. [3] Berlin-Brandenburg Institute of Advanced Biodiversity Research, Berlin, Germany. [4] VIB Center for Plant Systems Biology, Ghent, Belgium. [5] Ghent University, Department of Plant Biotechnology and Bioinformatics, Ghent, Belgium. [6] University of York, Department of Biology, York, UK. [7] Utrecht University, Department of Biology, Ecology and Biodiversity, Utrecht, The Netherlands. ✉email: r.dejonge@uu.nl; ville.friman@york.ac.uk; A.L.C.Jousset@uu.nl

Mutualistic interactions between multicellular hosts and their associated microbiota are important for the fitness of both parties[1–4]. However, while commonly observed in nature, direct evidence for the evolution of mutualism at both phenotypic and genotypic level is still limited[5–7]. The rhizosphere is a hotspot for mutualistic interactions between the plant and free-living microorganisms. For example, plants can preferentially interact with mutualistic microbes present in the indigenous species pool of the soil and disproportionally increase their relative abundances in the rhizosphere[8–10]. While such plant-mediated ecological filtering can rapidly change the relative abundances of mutualistic versus antagonistic species in the rhizosphere, it is less clear if plants can drive evolution of mutualism within species by increasing the fitness of emerging de novo mutualist genotypes. For example, even the most well-known plant mutualistic microbes, nitrogen-fixing rhizobia[5] and phosphorus-providing mycorrhizae[11], can be detrimental to the plant, suggesting that the interaction between a given pair of plant and microorganism varies naturally[12,13]. It is thus possible that plant-associated microbes might evolve along the parasitism–mutualism continuum in response to selection exerted by plants.

Beneficial symbioses between eukaryotic and prokaryotic organisms have evolved multiple times across the eukaryotic domain[14] and are considered as one of the major evolutionary transitions of life[15]. It has been suggested that the evolution of mutualism often requires two basic components: currency and mechanism of exchange of the currency[14]. In the context of plant–bacteria interactions, currency could be, for example, a root exudate, which can be taken up by bacteria. Similarly, bacteria might produce plant growth-promoting hormones such as auxin and gibberellins[16], which could be beneficial for plant growth. When the currency exchange between both parties is symmetrical, the selection is expected to favour the evolution of mutualism. Increased mutualistic dependence is then thought to evolve via reciprocal coevolution or via adaptation by one of the partners via selection on traits that are directly involved in the mutualistic interaction[15]. Currency exchange could also be asymmetrical, due to competition for shared limiting nutrients, such as iron[17], which could explain why certain plant–microbe interactions are antagonistic. Moreover, due to the open nature of the rhizosphere, free diffusion of plant-derived resources could select for increased levels of cheating where mutant bacterial genotypes take advantage of 'public goods' without contributing to the production of plant growth-promoting compounds[5,17]. As a result, mutualistic plant–microbe interactions might require additional enforcing from the plant[5] via sanctioning of cheating bacterial genotypes or by positively discriminating plant growth-promoting genotypes.

To assess whether plant–microbe mutualism can emerge as a consequence of plant-mediated effects, we apply an in vivo experimental evolution design[18], allowing the rhizosphere bacterium *Pseudomonas protegens* CHA0 to evolve on the roots of *Arabidopsis thaliana* in the absence of other microbes. This is achieved using sterile sand free of organic carbon as the growth substrate. As a result, bacterial survival and evolution is solely dependent on the presence of the plant, and the performance of evolved bacterial selection lines can be quantified in comparison with the ancestral bacterial strain. To set up the selection experiment, a clonal ancestral *P. protegens* bacterial population is inoculated on the roots of five independent *A. thaliana* Col-0 replicate plants (i.e., five plant replicate selection lines) and plants and bacteria grown in otherwise gnotobiotic conditions for a total of six plant growth cycles (each cycle lasting for 4 weeks). At the end of every growth cycle, the evolved bacterial populations are isolated and transferred to the rhizosphere of new sterile plants (Fig. S1). As a result, only bacteria are let to evolve, and the plant genotype is kept constant throughout the selection experiment. In

these experimental conditions, the initial plant–bacterium interaction is antagonistic: *A. thaliana* aboveground biomass is reduced in the presence of *P. protegens* CHA0 after one growth cycle ($F_{1, 8} = 45.4$, $P < 0.001$, Fig. 1a), and a likely cause for this is the production of diverse bioactive metabolites by CHA0[19] that can constrain plant growth[20]. To quantify changes in plant–bacterium interaction, 16 evolved bacterial colonies are randomly selected from each plant replicate selection line at the end of the second, fourth and sixth growth cycles, in addition to 16 randomly selected ancestral colonies (in total, 256 isolates). Each isolated colony is characterized phenotypically by measuring multiple key life-history traits, including growth on different carbon sources and media, tolerance to diverse abiotic and biotic stresses, production of several bioactive compounds and their ability to inhibit other microorganisms (Table S1). A subset of bacterial phenotypes is also subjected to full genome sequencing and characterized for their effects on plant growth in terms of root architecture, above and belowground biomasses, and activation of the root-specific transcription factor gene *MYB72* at the end of the selection experiment. The overall results illustrate that *P. protegens* rapidly diversifies into mutualist genotypes that have positive effects on the plant growth and competitive advantage over ancestral genotypes in the rhizosphere.

## Results

**Selection in the plant rhizosphere leads to bacterial phenotypic diversification and evolutionary transition towards mutualism.** To study the evolution of *P. protegens* CHA0 in the *A. thaliana* rhizosphere, we isolated a total of 240 evolved bacterial isolates from every second time point along with sixteen ancestral isolates (Supplementary Data 1) and used *K*-means clustering analysis to separate them into five distinct phenotypic groups based on several life-history traits (Fig. S2 and Table S2). The phenotypic groups were then given names that reflected key differences in their life-history traits and their mean effects on plant growth (Fig. 1 and Figs. S3 and S4). Evolved clones that clustered together with the ancestral strains were named as 'Ancestral-like' phenotype. Another phenotype similar to the ancestral strain, which only appeared momentarily before dropping below detection level, was named as 'Transient' phenotype (Figs. 1 and 2). A third phenotype that had clearly reduced abiotic stress tolerance ($F_{5, 248} = 40.8$, $P < 0.001$, Fig. S4) and increased ability to form a biofilm ($F_{5, 249} = 196.8$, $P < 0.001$, Fig. S4) was named as 'Stress-sensitive' phenotype. Agar plate assays were used to determine the effect of the evolved phenotypes on *A. thaliana* growth. While the 'Ancestral-like', 'Transient' and 'Stress-sensitive' phenotypes showed neutral effects on plant biomass relative to plant-only controls (shoot biomass, $F_{6, 26} = 8.01$, $P < 0.001$; root biomass, $F_{6, 26} = 2.84$, $P = 0.029$, Fig. 1), they had a negative effect on the plant root length (root length, $F_{6, 26} = 10.01$, $P < 0.001$, Fig. 1d) and caused a clear bleaching of plants indicative of reduced chlorophyll activity similar to the ancestral strain (the amount of green pixels, $F_{6, 26} = 5.90$, $P < 0.001$, Fig. 1f). These assays also revealed two novel phenotypes that showed positive effects on plant shoot and root biomasses (Fig. 1b, c) with comparable levels of plant 'greenness' to plant-only controls (Fig. 1f). These evolved phenotypes were therefore named as 'Mutualist 1' and 'Mutualist 2'as indicated by their plant growth-promoting activity.

The relative abundance of different phenotypic groups changed over time (Fig. 2a). The 'Ancestral-like' phenotypes persisted throughout the experiment even though they were substituted by evolved phenotypes in all plant selection lines (Fig. 2a). The evolutionary success of 'Transient' and 'Stress-sensitive' phenotypes was generally short-lived: 'Transient' phenotypes disappeared below the detection limit in all plant selection lines by the

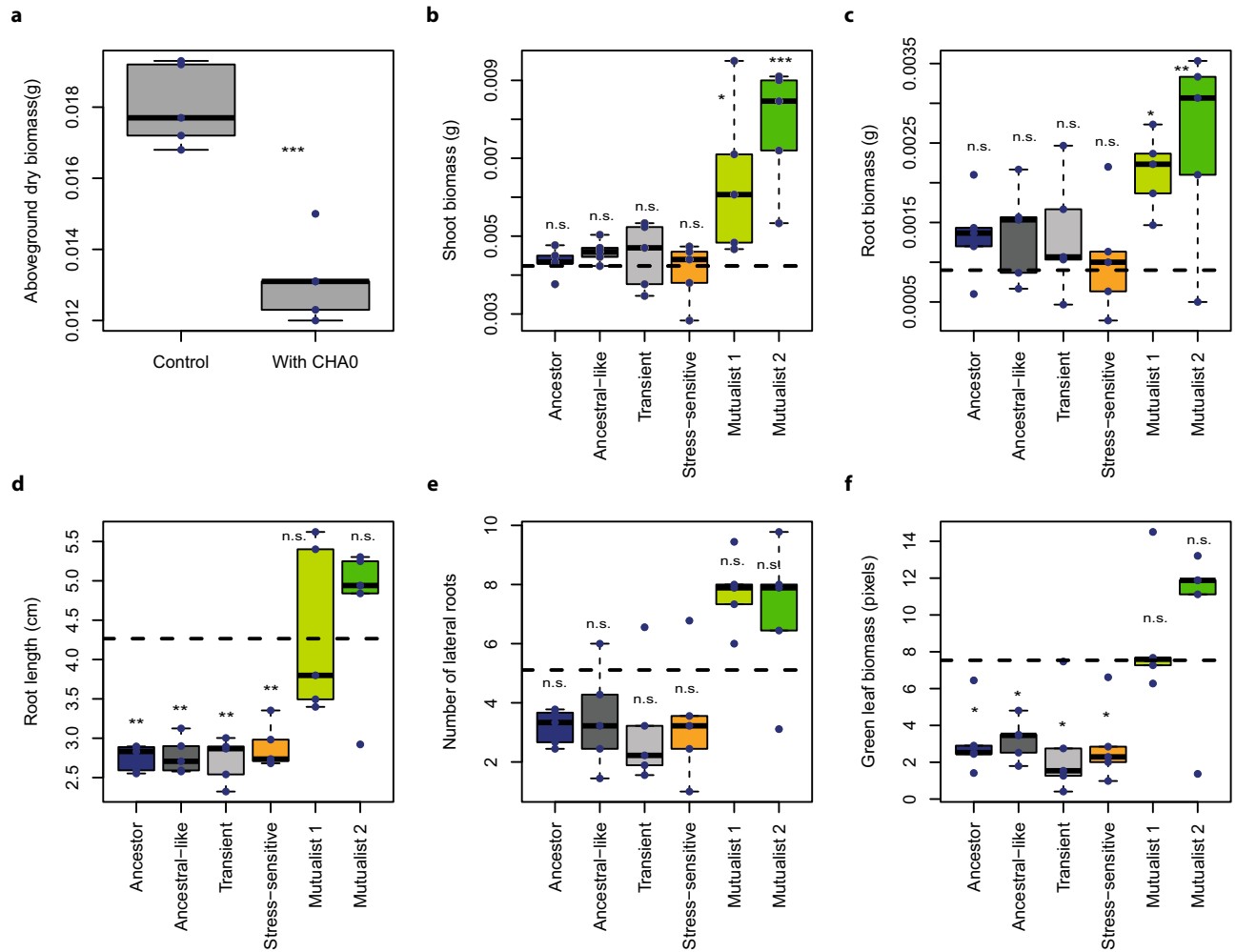

**Fig. 1 Evolution of bacterial mutualism in the rhizosphere of *Arabidopsis thaliana*.** Panel **a** shows the initially antagonistic effect of *Pseudomonas protegens* CHA0 on *A. thaliana* after one plant growth cycle in the sterile sand study system ($n = 5$; aboveground biomass ***$P = 0.0001$). Panels **b**–**f** compare the effects of ancestral and evolved *Pseudomonas protegens* CHA0 phenotypes on plant performance-related traits in a separate plant growth assays performed on agar plates at the end of the selection experiment ($n = 3$ for control and $n = 5$ for each evolved phenotype, see Table S2). Different panels show the shoot biomass in grams (**b**), root biomass in grams (**c**), number of lateral roots (**d**), root length in cm (**e**), and the amount of plant 'greenness' in terms of green-to-white pixel ratio (**f**) after 14 days of bacterial inoculation (Supplementary Data 2; blue dashed horizontal lines show the non-inoculated control plants). Bacterial phenotype groups are displayed in different colours (black: ancestor; dark grey: ancestral-like; light grey: transient; orange: stress-sensitive, light green: mutualist 1 and dark green: mutualist 2) and were classified and named based on *K*-means clustering (Fig. S1) using 14 phenotypic traits linked to growth, stress tolerance, production of bioactive compounds and antimicrobial activity (Table S1). All boxplots show median (centre line), interquartile range (25–75%) and whiskers that extend 1.5 times the interquartile range overlaid with a scatter plot showing independent replicates. Statistical testing in all panels was carried out using one-way ANOVA, and asterisks above plots indicate significant differences between control and bacteria-treated plants (*$P = 0.05$, **$P = 0.01$, ***$P = 0.001$; n.s. = non-significant). Data for all panels are provided in the Source Data file.

end of the sixth growth cycle (Fig. 2a), while the 'Stress-sensitive' phenotypes emerged only in three selection lines and survived until the end of the experiment only in one of the selection lines (Fig. 2a). In contrast, the frequency of mutualistic phenotypes increased in four out of five plant selection lines throughout the experiment, while one selection line became dominated by 'Ancestral-like' and 'Stress-sensitive' phenotypes (Fig. 2a).

An aggregated 'plant performance' index summarising the effects of each bacterial isolate on both aboveground and belowground plant growth traits (Fig. 2b, PC1 of multivariate analysis) was used to explore if reduced antagonism towards the plant was associated with improved bacterial growth indicative of the evolution of a reciprocally beneficial mutualistic interaction. We found a significant positive correlation between plant performance index and bacterial phenotype abundance per plant ($F_{1, 28} = 8.01$, $P < 0.001$, Fig. 2c). Specifically, both mutualistic

phenotypes reached higher abundances in the plant rhizosphere compared to other phenotypes (bacterial cells per plant). This indicates that reduced bacterial antagonism towards the plant was coupled with improved growth in the rhizosphere, which could also explain why mutualists became the dominant phenotypes in four out of five plant selection lines during the selection experiment (reaching up to 94% relative abundance, Fig. 2a). In support for this, a similar positive correlation was observed between the degree of plant performance of each phenotype measured in separate plant growth assays and their relative abundance in diversified rhizosphere populations at the end of the selection experiment ($F_{1, 23} = 4.37$, $P = 0.048$, Fig. S5). Together, these results demonstrate that the evolution of plant-growth promotion was accompanied with increased bacterial fitness, indicative of a mutualistic interaction where each species had a net benefit. As this evolutionary transition was observed in

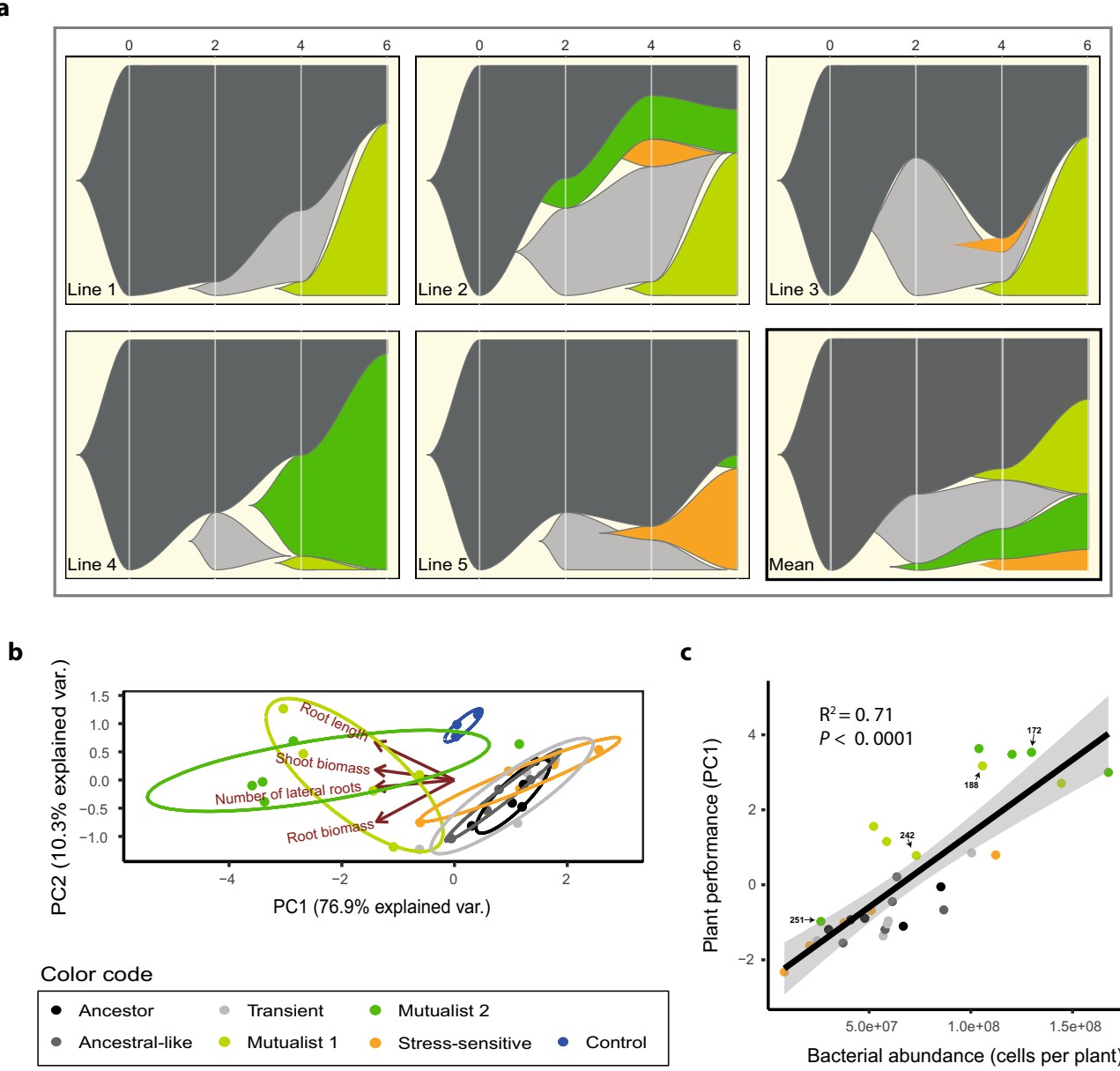

**Fig. 2 Temporal changes in bacterial phenotypes during the selection experiment and positive correlation between evolved bacteria and plant growth.**
Panels in **a** show the dynamics of five bacterial phenotype groups across five plant replicate lines and the overall mean pattern during six growth cycles (6 months). The *x*-axis shows the plant growth cycle (0: ancestral bacterium) and the *y*-axis shows the relative abundance of each bacterial phenotype. Panel **b** shows a principal component analysis (PCA) for five representative bacterial isolates from each evolved phenotype group in addition to ancestor isolates (see Table S2) based on their plant growth-related traits. The negative PC1 values of each isolate were extracted and combined to a 'Plant performance' index, which included bacterial effects on shoot biomass, root biomass and root architecture explaining 76.9% of the total variation in plant growth. Panel **c** shows a positive correlation between 'Plant performance' and bacterial abundance on the plant roots at the end of the fitness assays; the black line and grey area indicate the linear regressions with 95% confidence intervals, respectively ($n = 30$, biologically independent isolates, see Table S2; $P = 4.296e-09$). In all panels, bacterial phenotype groups are displayed on different colours (black: ancestor; dark grey: ancestral-like; light grey: transient; orange: stress-sensitive, light green: mutualist 1 and dark green: mutualist 2). The sample IDs of four isolates from the two mutualistic phenotype groups are highlighted on labels. Data for all panels are provided in the Source Data file.

parallel in four out of five selection lines, it was likely driven by deterministic processes such as selection exerted by the plant or bacterial competitive superiority instead of random genetic drift due to bottlenecking between plant growth cycles.

**Evolution of mutualism is linked to improved resource catabolism and tolerance to plant-secreted antimicrobials.** For stable mutualism to evolve, plants would need to provide the

evolved mutualists a 'currency' that could not be accessed by the other phenotypes or employ some form of 'sanctioning' to constrain the growth of non-mutualist phenotypes. To study this, we first compared differences in the evolved phenotypes' ability to use a range of carbon sources that are typically found in *A. thaliana* root exudates[21], and which could have selectively preferred the growth of mutualist phenotypes. Second, we compared the evolved phenotypes' tolerance to scopoletin, which is an

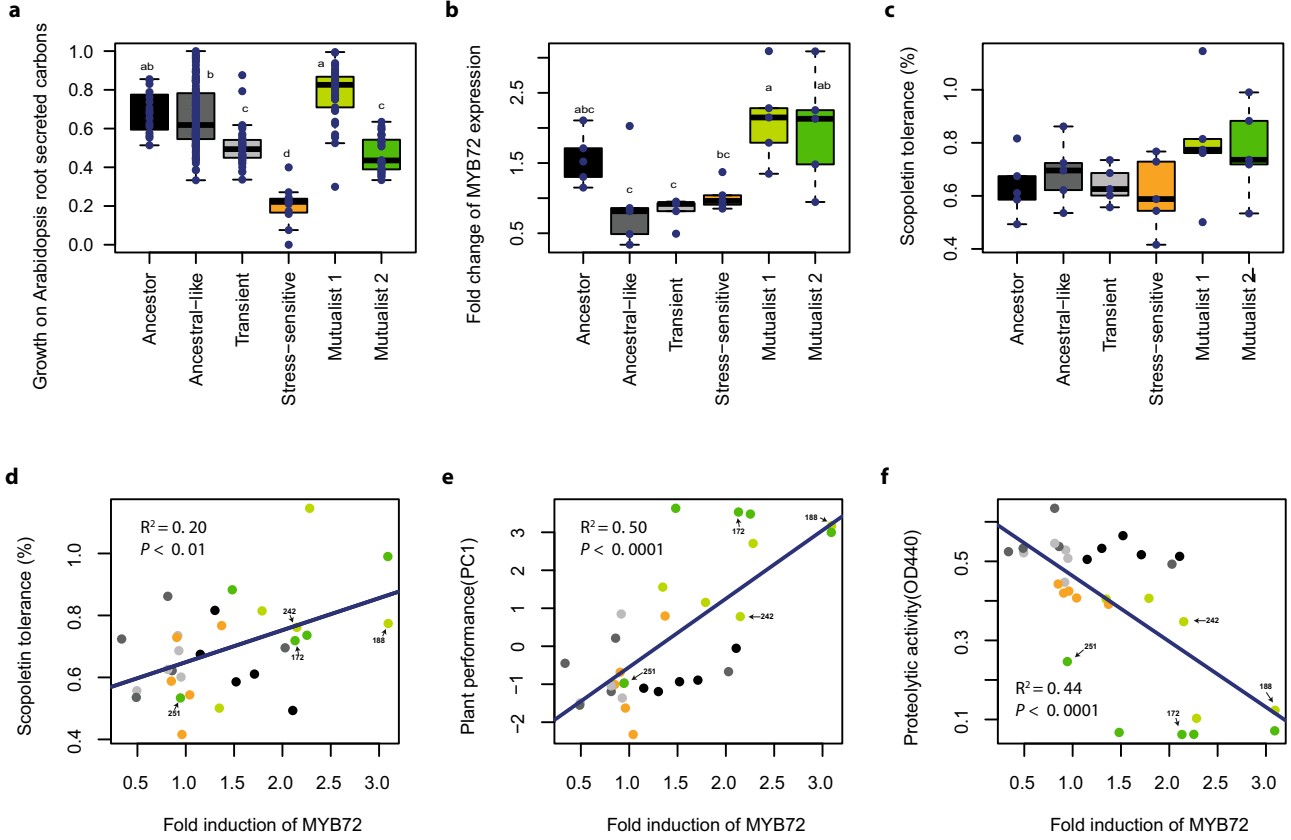

**Fig. 3 Selection mechanisms favouring the increase in the relative abundance of mutualists in the rhizosphere of *Arabidopsis thaliana*.** Panel **a** shows the growth of ancestor and evolved *Pseudomonas protegens* CHA0 phenotypes on carbons typically secreted by *A. thaliana* (14 most dominant carbons analysed as a combined index based on normalised first principal component PC1, which explained 83.9% of total variation). In total, 256 isolates were characterized including ancestral ($n = 16$), 'Ancestral-like' ($n = 119$), 'Transient' ($n = 41$), 'Stress-sensitive' ($n = 11$), 'Mutualist 1' (37), and 'Mutualist 2' ($n = 31$) phenotypes. Panel **b** shows the effect of ancestor and evolved *P. protegens* CHA0 phenotypes on the expression of *MYB72* (transcription factor responsible for scopoletin production) in the roots of a GUS *A. thaliana* reporter line (based on the quantification of GUS staining of the roots, Fig. S6). Panel **c** shows the relative growth of ancestor and evolved *P. protegens* CHA0 phenotypes in the presence of the plant-secreted scopoletin antimicrobial at 2 mM concentration after 72 h of incubation relative to no-scopoletin control. Panel **d** shows a positive relationship between *MYB72* expression (fold induction; *x*-axis) and scopoletin tolerance (*y*-axis) for all tested isolates. Panel **e** shows a positive relationship between *MYB72* expression (fold induction; *x*-axis) and plant performance (*y*-axis) for all tested isolates. Panel **f** shows a negative relationship between *MYB72* expression (fold induction; *x*-axis) and proteolytic activity (*y*-axis) for all tested isolates. The sample IDs of four isolates from the two mutualistic phenotype groups are highlighted on labels (see Table S2) in panels **d**–**f**. Panels **b**–**f** include five representative bacterial isolates ($n = 5$, biologically independent isolates) from each phenotype in addition to the ancestor (each replicate line represented; see Table S2). In all panels, bacterial phenotype groups are displayed on different colours (black: ancestor; dark grey: ancestral-like; light grey: transient; orange: stress-sensitive, light green: Mutualist 1 and dark green: Mutualist 2). All boxplots show median (centre line), interquartile range (25–75%) and whiskers that extended 1.5 times the interquartile range overlaid with a scatter plot showing independent replicates. Statistical testing in panels **a**–**c** was carried out using one-way ANOVA followed by Tukey's multiple comparison test ($\alpha = 0.05$; different lowercase letters indicate significant differences). Panels **d**–**f** show linear regression (black line) and Pearson correlations fitted over all biologically independent isolates ($n = 30$). Data for all panels are provided in the Source Data file.

antimicrobial secreted by plant roots known to modulate the composition of the root-associated microbial community by favouring more tolerant bacterial taxa[22,23]. We found that the 'Mutualist 1' phenotypes showed an improved ability to grow on various carbon sources typical for *A. thaliana* root exudates compared to the other phenotypes (PC1 of multivariate analysis, $F_{5, 249} = 46.67$, $P < 0.001$, Fig. 3a). Moreover, 'Transient', 'Stress-sensitive' and 'Mutualist 2' phenotypes showed reduced growth on carbon sources relative to 'Ancestral-like' phenotypes indicative of competitive disadvantage (Fig. 3a). These results suggest that 'Mutualist 1' phenotypes potentially evolved to be better at competing for plant-derived root exudates, which could have increased their abundance relative to other phenotypes.

To explore the potential significance of the antimicrobial scopoletin, we used a GUS reporter assay to determine how different bacterial phenotypes affected the expression of the plant

root-specific transcription factor gene *MYB72*[24], which encodes a known positive regulator of scopoletin biosynthesis[22]. We found that plants inoculated with 'Mutualist 1' and 'Mutualist 2' phenotypes retained high GUS activity, which was comparable to the ancestor ($F_{5, 24} = 5.6$, $P < 0.01$, Fig. 3b and Fig. S6). However, the other evolved phenotypes induced a reduced GUS activity in plants relative to the mutualists (Fig. 3b and Fig. S6). While the 'Mutualist 1' and 'Mutualist 2' phenotype groups included strains that showed very high tolerance to scopoletin, their mean tolerance did not significantly differ from other phenotype groups ($F_{5, 24} = 1.76$, $P = 0.16$, Fig. 3c). However, a significant positive correlation was observed between the induction of *MYB72* in *A. thaliana* roots and phenotypes' tolerance to scopoletin ($F_{1, 28} = 8.29$, $P < 0.01$, Fig. 3d). This suggest that both the activation of scopoletin production and the scopoletin tolerance were potentially under co-selection as the only mutualistic phenotypes

showing high scopoletin tolerance were those able to upregulate the expression of *MYB72*, a gene positively regulating scopoletin production by the plant. Such mechanism could have ensured positive selection for scopoletin-inducing, scopoletin-tolerant mutualists relative to other evolved phenotypes. We also found that *MYB72* induction was positively correlated with plant growth (i.e., plant performance, Fig. 3e) and negatively correlated with phytotoxic compounds (i.e., exoprotease production, Fig. 3f). Moreover, scopoletin tolerance was positively correlated with both plant performance (Fig. S6A) and bacterial fitness on the plant roots (Fig. S7A). Together, these results suggest that both mutualists had positive effects on the plant growth, and that plants could positively discriminate mutualist phenotypes by providing specific 'nutrient' and 'tolerance' niches, creating a positive feedback loop.

**Mutualistic phenotypes had mutations in genes encoding the GacS/GacA two-component regulatory system.** To gain insights into the genetic mechanisms underlying the evolution of mutualism, we performed whole-genome re-sequencing of 30 isolates followed by reference-based identification of single-nucleotide polymorphisms (SNPs) and small and large insertions or deletions (INDELs). Because bacterial populations consisted only of one bacterial species, and as no plasmids or large genomic islands have been detected in the *P. protegens* CHA0 genome, we excluded the potential role of mobile genetic elements in explaining the phenotypic variation[25]. These analyses revealed that different evolved bacterial phenotypes were associated with relatively few mutations in global regulator genes (Fig. 4), underpinning their central role in bacterial adaptation[26,27]. While only a few non-parallel mutations were observed in case of 'Ancestral-like' and 'Transient' phenotypes (Table S2), all but two mutualistic isolates (8/10) harboured mutations in genes encoding the GacS/GacA two-component system, which regulates secondary metabolism alongside many other aspects of bacterial physiology[28]. Despite a high level of parallelism, a variety of different mutations were observed within the *gacS* and *gacA* genes. Three *gacS/gacA* mutations were unique to 'Mutualist 1' isolates, and specifically associated with an uncharacterized N-terminal histidine kinase domain in GacS (G27D) and the response regulatory domain of GacA (G97S; D49Y, Fig. 4 and Table S2). Three unique 'Mutualist 2' mutations were found inside the *gacA* coding region (E38X, D54Y and Y183S, Fig. 4 and Table S2). Mutations upstream the transcription start site (−40 nucleotides), potentially representing a promoter binding site, were detected within both mutualists (Table S2). The conserved phosphate-accepting aspartate 54 (D54) residue is important for phospho-relay initiated by the sensor kinase GacS, and mutations of this residue are associated with complete loss-of-function[29–31]. Aspartate 49 (D49) is another conserved residue in the vicinity of D54, and the *gacA* (49Y) allele has previously been reported to be associated with a partial reduction in GacA activity[31]. The other mutations in *gacA* are novel and conceivably have a significant impact on GacA activity as they result in a severely truncated protein (E38X) or are located within the third recognition helix of the LuxR-like tetra-helical helix-turn-helix (HTH) domain, which is known to make most of the DNA contacts (Y183S)[32]. In line with the predicted effects of the mutations, 'Mutualist 1' isolates retained part of the GacS/GacA-mediated traits, while 'Mutualist 2' isolates showed a severe to complete disruption of extracellular proteolytic and antifungal activity (Fig. S3). Interestingly, 'Mutualist 1' isolates with mutations in *gacS* (G27D) and *gacA* (D49Y) both showed a lower plant performance compared to other *gac* mutants. Moreover, a single mutation in the *fleQ* (R320Q) gene was found in mutualist ID251 (line 5), with no

clear link with plant performance (Fig. 4a), while a mutation in the *accC* (E413K) gene was identified in certain mutualistic isolates with positive effect on plant performance (Fig. 4a). The gene *accC* encodes the biotin decarboxylase subunit of the acetyl coenzyme A carboxylase complex involved in fatty acid biosynthesis and its role on bacterial physiology and bacteria–plant interactions should be validated in the future. Together, these differences between individual mutations are likely to explain the observed variation in life-history traits (including plant performance) within and between the 'Mutualist 1' and 'Mutualist 2' groups.

Interestingly, while mutualists emerged in all plant replicate selection lines, they did not become the dominant phenotype in selection line 5. Instead, this selection line became dominated by 'Stress-sensitive' bacteria (Figs. 1 and 4), which also transiently appeared in two other plant selection lines. Genetically, the 'Stress-sensitive' phenotype was mainly associated with mutations in the *rsmY* region in *P. protegens* CHA0[33] including non-synonymous mutations in *nlpD* (Q197P) and *rpoS* (Q65X). *NlpD* encodes a lipoprotein predicted to play a role in cell wall formation and cell separation and is found immediately upstream of *rpoS*. The *rpoS* gene encodes sigma factor $\sigma^{38}$, which mediates general stress resistance[34,35], downregulates the biosynthesis of antagonistic secondary metabolites[36] and is involved in biofilm formation[37] in several bacterial taxa. In line with this, we found that 'Stress-sensitive' phenotypes were able to form high amounts of biofilm in vitro (Fig. S4), which may have supported more efficient root colonization[38], explaining their dominance. More efficient root colonization could have further initiated a strong priority effect[39,40], potentially constraining the subsequent emergence of mutualistic bacterial phenotypes within the time frame of the experiment. Mutations in *nlpD* has been linked to alterations in colony morphology, possibly through changes in the expression of *rpoS* as *nlpD* has been found to exert a promoter function on *rpoS* in *Escherichia coli*[41] and *Pseudomonas aeruginosa*[42]. One of the sequenced 'Stress-sensitive' clones had also a mutation in a TetR-family transcriptional regulator (*tetR*). Interestingly, this mutation had a tentative epistatic effect on *rpoS* as indicated by the relatively stronger, but non-significant, effect of the *rpoS·tetR* double mutant on plant performance. Notably, the TetR-family regulator PsrA was previously reported to regulate the expression of *rpoS* in *Pseudomonas* spp.[42], further strengthening these observations. Together, these results show that while plant selection can lead to a high level of parallel evolution both at the phenotypic and molecular level, alternative evolutionary trajectories are also possible.

**The fitness benefits of GacS/GacA mutations are specific to the rhizosphere environment.** In order to assess whether the observed mutations specifically conferred an advantage in the rhizosphere environment, we compared the fitness of evolved mutualists on plant roots and in liquid growth culture media. To this end, the fitness of two evolved *gacA* (Mutualists 1 and 2; ID 242 and ID 220, respectively, Table S2), and one *gacS* genotype (Mutualist 1, ID 222, Table S2) was compared relative to their direct ancestral genotypes without *gac* mutations (ID 133, ID 28 and ID 66, respectively, Table S2) within the same plant selection lines. Fitness was determined as the relative competitive fitness in direct pairwise competitions as a deviation from the initial 1:1 ancestor-to-successor ratio in vivo on *A. thaliana* roots and in vitro in Kings' B (KB), lysogeny broth (LB) and tryptic soy broth (TSB) growth media. Post-competitive genotype ratios were determined using PCR-based high-resolution melting profile (RQ-HRM) analysis (see 'Methods' and Supplementary Fig 7). It was found that all three *gacS/gacA* mutants had a higher fitness in

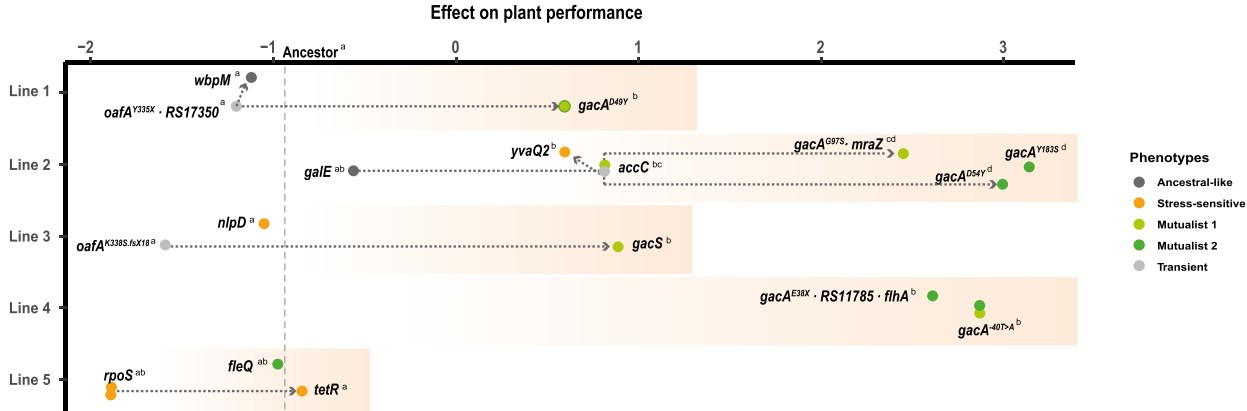

**Fig. 4 Genetic basis of bacterial evolution in the rhizosphere of *Arabidopsis thaliana*.** Panel **a** shows clear parallel evolution between four out of five plant replicate selection lines based on re-sequencing of 25 evolved and five ancestor isolates used in the phenotypic assays. Filled dots represent isolates with novel mutations (present in 18/25 evolved isolates). The seven isolates without mutations, or only synonymous mutations, are not included. The x-axis shows a combined index of 'Plant performance' relative to non-inoculated control plants (values on the x-axis indicate positive and negative effects on the plant and the y-axis shows the five independent plant replicate selection lines). The effect of the ancestral bacterial genotype on plant performance is shown as a vertical dashed line. Statistical testing in panel **a** was carried out using one-way ANOVA (each line analysed separately). The different letters on the top right of each genotype indicate significant differences based on a Tukey's HSD test ($\alpha = 0.05$; each line analysed separately, $n = 3$). Bacterial phenotype groups are displayed on different colours (black: ancestor; dark grey: ancestral-like; light grey: transient; orange: stress-sensitive, light green: Mutualist 1 and dark green: Mutualist 2) and the accumulation of mutations within replicate lines are shown with connected dashed arrows. Panel **b** table lists unique mutations linked with evolved bacterial phenotypes. Successive mutations that appeared within the same genetic background are shown after the indent. Notably, these additional mutations did not affect the bacterial phenotypes (see Table S3 for a more detailed description of the mutations). Data for all panels are provided in the Source Data file.

the rhizosphere relative to their direct ancestral genotypes without *gac* mutations ($F_{3, 32} = 10.03$, $P < 0.001$, Fig. 5). Interestingly, this advantage was smaller for one of the *gacA* mutants (ID 220, $F_{2, 6} = 15.35$, $P = 0.004$, Fig. 5) likely because its direct ancestor already showed mutualistic behaviour due to a mutation in the *accC* gene, reducing the relative benefit of the *gacA* mutation within this lineage (Table S2). While the fitness benefits of *gac* mutations were mainly observed in the rhizosphere, two *gac* mutants showed improved competitive fitness in KB media indicative of general metabolic adaptations (genotype × measurement environment: $F_{6, 24} = 13.02$, $P < 0.001$; genotype comparisons in KB media: $F_{2, 6} = 162.6$, $P < 0.001$). Together, these results confirm that the genetic changes underlying the evolution

of bacterial mutualism were mainly beneficial in the presence of plants.

## Discussion

Even though beneficial plant–microbe interactions are widely documented, their evolutionary origin is less well understood. Here we studied how an initially antagonistic relationship between the *P. protegens* CHA0 bacterium and its host plant, *A. thaliana*, changed during prolonged selection over six plant growth cycles (6 months). While several studies have previously reported beneficial effects of CHA0 on plant growth in natural soils, it initially showed antagonism towards the plant in our

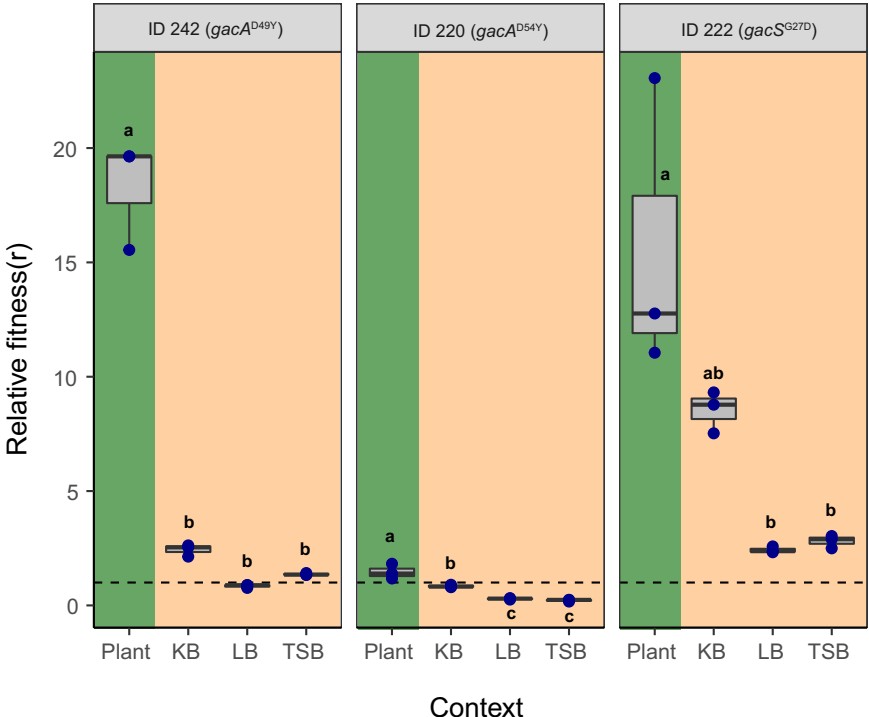

**Fig. 5 Competitive fitness of *gac* mutants relative to their direct ancestors in the rhizosphere and in in vitro culture media.** The *gac* mutants' relative fitness (*r*) was calculated based on the deviation from the initial 1:1 genotype ratio (dashed line) after direct competition in different environments. Fitness values above the dashed line indicate a higher competitive advantage of *gac* mutants relative to their ancestral genotypes without *gac* mutations (Table S2), whereas values below the dashed line denote for decreased competitive ability of evolved *gac* mutants. In all panels, green and beige backgrounds denote competition assays conducted in the rhizosphere and in standard culture media, respectively. All boxplots show median (centre line), interquartile range (25–75%) and whiskers that extended 1.5 times the interquartile range overlaid with a scatter plot showing independent replicates ($n =$ 3). Different small letters above the boxplots represent significant differences in relative fitness (*r*) between growth conditions for each mutant (one-way ANOVA, Tukey's HSD test, $\alpha = 0.05$). Data for all panels are provided in the Source Data file.

experimental conditions due to production of phytotoxic compounds[20] (Fig. 1). Crucially, this interaction rapidly evolved during the experiment as bacterial mutualists that had positive effects on the plant growth and relatively higher competitive advantage in the rhizosphere, emerged and swept through replicate bacterial populations.

After the initial bacterial diversification in the rhizosphere, changes in bacterial genotype frequencies could have been driven by bacterial competitive differences. In this case, mutualism could have risen as a simple by-product of bacterial adaptation where competitive superiority was coupled with plant-growth promotion. In support, we found that both mutualists showed reduced secretion of extracellular proteolytic enzymes (Fig. S3A), which was positively correlated with improved plant performance (Fig. S6E). As many metabolites produced by *Pseudomonas* spp. are costly to produce[20], such change could have improved the growth of mutualistic bacteria, which was clearly the case with 'Mutualist 1' phenotypes. Interestingly, 'Mutualist 1' ability to grow on model root exudates was also positively correlated with *MYB72* gene induction, which was associated with a positive effect on plant growth. As a result, increased *MYB72* induction could have created a positive feedback loop between plant growth-promotion and 'Mutualist 1' growth on plant root exudates. While evolved 'Mutualist 2' phenotypes also showed lowered proteolytic activity, they did not exert improved growth on tested root exudates. Instead, 'Mutualist 2' phenotypes showed relatively greater increase in their tolerance to the plant-derived antimicrobial scopoletin. Crucially, scopoletin tolerance was also positively coupled with *MYB72* gene induction, which is known to be important for scopoletin production and secretion by *A.*

*thaliana*[22]. As a result, plant growth-promotion via *MYB72* induction likely improved 'Mutualist 2' competitiveness by leading to increased scopoletin production, which was relatively more harmful to other bacterial phenotypes. The importance of plant selection in driving changes in genotype frequencies was further demonstrated by direct competition assays, where the fitness benefits of mutualists were clearest when measured in the presence of plants. Together, these findings suggest that bacterial mutualism was driven by evolution of reduced phytotoxicity and increased *MYB72* induction, which increased both plant growth and mutualists' competitive ability in the plant rhizosphere. Increased *MYB72* induction thus likely favoured both types of mutualists, potentially explaining their coexistence within certain plant selection lines (2 and 4).

At the genetic level, evolution of mutualism could be achieved with only a single or a few successive mutations involving mainly global regulators[30]. This result shows an interesting parallel with recent work demonstrating that the loss of a few virulence traits can turn a pathogen into a beneficial symbiont[7]. Evolution of mutualism was also linked with clear phenotypic and genotypic bacterial diversification, which has previously been observed in aquatic[43] and soil[44] microcosms in response to spatial heterogeneity. Here we show that such bacterial diversification can also be enforced by plant selection as evidenced by direct competition assays where fitness benefit observed in the rhizosphere was reduced or undetectable in lab media in vitro. Parallel mutations in both *gacS* and *gacA* genes suggests that these genes played an essential role in regulating both bacterial competition and interaction with the plant. For example, it is well established that this two-component system regulates the production of phytotoxic

compounds, such as exoprotease, 2,4-DAPG, hydrogen cyanide and biosurfactants[20,45,46]. As a result, mutations in *gacS* and *gacA* genes likely explain observed pleiotropic effects between antagonistic activity[29–31] and plant performance. These mutations could also have a positive effect on bacterial competitive ability via reduced metabolic costs[47], and *gacA* mutations have been shown to affect 15% of annotated genes in *P. aeruginosa* M18 including genes involved in the motility, secretion systems and assimilation and metabolism of phosphorus, sulphur, and nitrogen[48]. Even though the role of *gac* mutations is not known in the case of plant *MYB72* induction, *gacA* has been shown to be important for the full virulence of the plant pathogen *Pseudomonas syringae*[49]. In this regard, our results are in line with previous studies demonstrating that *gacS* and *gacA* mutations selected for less antagonistic (or virulent) *P. protegens* CHA0 genotypes, similar to pathogenic *Pseudomonas* species. While more research on the underlying molecular mechanisms is required, our findings suggest that the GacS/GacA two-component system could be an important pleiotropic switch for enabling evolution of mutualistic interaction with plants.

Interestingly, we observed a contrasting evolutionary outcome in one of the five selection lines where 'Stress-sensitive' genotypes were able to become dominant alongside with 'Ancestral-like' genotypes potentially due to their enhanced ability to form biofilm and plant root colonization. Interestingly, none of the phenotypes was able to reach fixation in the rhizosphere. One possibility for this is that the experiment was not long enough for the selective sweeps to drive beneficial mutations into fixation, or for the mutualist genotypes to emerge in all selection lines. Alternatively, it is possible that multiple phenotypes were able to coexist due to negative frequency dependent selection or because they occupied different spatial niches as seen in heterogenous soil environments[44,50]. These hypotheses could be studied directly in the future using fluorescent microscopy and tagged strains to observe diversification and genotype fluctuations in the rhizosphere both in space and time.

In summary, our results show that in addition to recruiting beneficial bacteria from multi-species microbial communities[8,9,51], plants could also change the functioning of its associated microbiota by creating strong selection for de novo evolution of mutualistic bacterial genotypes. Steering bacterial evolution in the rhizosphere could thus offer plants a shortcut to improve their fitness without evolving themselves[52–54]. Future work should focus on validating our findings in more complex microbial communities where bacterial diversification could also be affected by interactions with other microbes. Moreover, it would be interesting to test if mutualism can evolve from initially neutral interaction and if plants can coevolve in response to rhizosphere bacteria. In conclusion, our results call for eco-evolutionary management of plant–microbe interactions in agriculture by demonstrating that plant-associated bacteria can rapidly evolve along the parasitism–mutualism continuum within a few plant growth cycles.

## Methods

**Bacterial strain and growth conditions**. We used *Pseudomonas protegens* (formerly *Pseudomonas fluorescens*)[55] CHA0 as a model strain, which was initially isolated from tobacco roots[56]. The strain was chromosomally tagged with GFP and a kanamycin resistance cassette to enable specific tracking of the strain and detection of contaminations[19]. This bacterium has the genetic potential to produce various bioactive metabolites, including the plant hormone indole-3-acetic acid (IAA), antimicrobial compounds and lytic enzymes[25]. Prior to the experiment, bacteria were grown for 48 h on a King's medium B[57] (KB) agar plate supplemented with 50 µg ml$^{-1}$ kanamycin, a single colony was randomly picked and grown for 12 h in KB at 28 °C with agitation. The cell culture was then washed for three times in 10 mM MgSO$_4$ and adjusted to 10$^7$ cells ml$^{-1}$ and used as inoculant for all plants. This inoculant was also stored at −80 °C as frozen ancestral stock, from which 'Ancestor' isolates were picked in later experiments.

**Host plant and growth conditions**. We used *Arabidopsis thaliana* ecotype Col-0 as a model host plant. Surface-sterilized seeds were first sown in Petri dishes with agar-solidified (1.5% agar (w/v)) modified Hoagland's medium: (KNO$_3$ (3 mM), MgSO$_4$ (0.5 mM), CaCl$_2$ (1.5 mM), K$_2$SO$_4$ (1.5 mM), NaH$_2$PO$_4$ (1.5 mM), H$_3$BO$_3$ (25 µM), MnSO$_4$ (1 µM), ZnSO$_4$ (0.5 µM), (NH$_4$)$_6$Mo$_7$O$_{24}$ (0.05 µM), CuSO$_4$ (0.3 µM), MES (2.5 mM) and 50 µM Fe(III)EDTA, pH = 5.8) and stratified for 2 days at 4 °C after Petri dishes were positioned vertically and transferred to a growth chamber (20 °C, 10 h light/14 h dark, light intensity 100 µmol m$^{-2}$ s$^{-1}$). After 2 weeks of incubation, two seedlings were transferred to closed and sterile ECO2 boxes (http://www.eco2box.com/ov80xxl_nl.htm) for selection experiment. The ECO2 boxes were filled with 260 g of dry, carbon-free silver sand that was previously washed with MilliQ water to remove dissolvable chemical elements and heated to 550 °C for 24 h to remove remaining organic material. Prior to transplantation the sand was amended with 13 ml of modified Hoagland medium.

**Design of the selection experiment**. The selection experiment was conducted in a gnotobiotic system to remove confounding effects that may emerge as a result of competitive interactions with other microorganisms, and to place the focus on plant-mediated selective pressures. Moreover, we allowed only the bacteria to evolve during the experiment and used new clonal plants at every bacterial transfer. We set up five independent plant–bacterium replicate lines, which were grown for six independent plant growth cycles (see Fig. S1 for an overview of the experimental design). The experiment was started by inoculating 10$^6$ cells of the stock *P. protegens* CHA0 culture (from here on abbreviated as 'ancestor') into the rhizosphere of 2-week-old *A. thaliana* seedlings growing in sterile silver sand within ECO2 boxes (two plants per replicate selection line). Inoculated plants were then grown for 4 weeks (20 °C, 10 h light/14 h dark, light intensity 100 µmol m$^{-2}$ s$^{-1}$) after which the plant growth cycle was terminated and root-associated bacteria were harvested by placing the roots of both plants into a 1.5 ml Eppendorf tubes filled with 1 ml 10 mM MgSO$_4$ and two glass beads. Rhizosphere bacteria were suspended into the liquid using a TissueLyser II at a frequency of 20 s$^{-1}$ for 1 min after which bacterial cell densities were determined using flow cytometry (BD Accuri™ C6 Plus, thresholds for FSC: 2000, SSC: 8000). After this, 10$^6$ cells were inoculated to the rhizosphere of new *A. thaliana* plants to initiate the next plant growth cycle. Possible contaminations were checked by plating the suspension on 3 g l$^{-1}$ tryptic soy agar (TSA) plates and it was verified that all colonies carried the *GFP* marker gene, as observed under UV light.

**Bacterial life-history traits measurements**. Individual bacterial colonies were isolated from all replicate plant selection lines for life-history measurements at the end of the second, fourth and sixth plant growth cycle by dilution plating the rhizosphere suspension on 3 g l$^{-1}$ TSA plates. After incubation at 28 °C for 24 h, 16 colonies were randomly picked from each replicate selection lines resulting in a total of 240 evolved and 16 ancestral colonies. All these colonies were characterized for a set of key bacterial life-history traits representative of growth, stress resistance and traits linked with plant–microbe interactions.

a. *Bacterial growth yield in KB medium*
   All the bacterial isolates were grown in 96-well plates with 160 µl 1/3 strength liquid KB, at 20 °C without shaking. Bacterial yield was determined as the maximum optical density at 600 nm after 3 days of growth using a spectrophotometer (SPECTROstar Nano).

b. *Bacterial stress resistance*
   We measured bacterial resistance to a range of different stresses using various 96-well microplate assays. Abiotic stress resistance was determined by growing bacteria in 160 µl of 1 g l$^{-1}$ TSB containing 0.0025% H$_2$O$_2$ (oxidative stress), 15% polyethylene glycol (PEG)−6000 (water potential stress) or 2% NaCl (salt stress). We used resistance to antibiotics commonly produced by rhizosphere microorganisms as indicator of biotic stress resistance. Antibiotic resistance was tested in 160 µl of 1 g l$^{-1}$ TSB supplemented with 1 µg ml$^{-1}$ streptomycin, 1 µg ml$^{-1}$ tetracycline, or 5 µg ml$^{-1}$ penicillin, respectively. Bacterial growth were determined after 3 days of growth at 20 °C without shaking as optical density at 600 nm.

c. *Traits linked with plant–microbe interactions*
   *P. protegens* CHA0 harbours several traits that are linked to plant growth including production of antibiotics and plant hormones. To assess these traits, we grew each bacterial colony in 96-well plates containing 160 µl of 1/3 strength liquid KB per well at 20 °C without shaking for 72 h. Cell-free supernatants were obtained by filter sterilization (0.22 µm) using Multiscreen HTS 96-well filtration plates (1000 × *g*, 30 min), which were used to measure the production of the plant hormone auxin (Indole-3-acetic acid (IAA)), iron-chelating siderophores and proteolytic activity. Furthermore, we also measured antifungal and antibacterial activity of all colonies.

*IAA detection*. The production of the plant hormone auxin was determined with a colorimetric test[58]. Briefly, 30 µl *P. protegens* CHA0 cell-free filtrate was incubated with 30 µl R1 reagent (12 g l$^{-1}$ FeCl$_3$, 7.9 M H$_2$SO$_4$) for 12 h in the dark and optical density read at 530 nm of the colorimetric complex was used as a measurement of IAA concentration.

*Siderophore activity*. Iron-chelating ability was measured as a proxy for siderophore production[59]. To this end, 100 μl of *P. protegens* CHA0 cell-free filtrate was mixed with 100 μl of modified CAS solution (with 0.15 mM FeCl3) and optical density read at 630 nm after 3 h of incubation was used as a proxy of siderophore production. The iron-chelating ability was calculated based on the standard curve based on modified CAS assay solution with a range of iron concentration (0, 0.0015, 0.003, 0.006, 0.009, 0.012, 0.015 mM FeCl3).

*Proteolytic activity*. The proteolytic activity assay we used was adapted from Smeltzer et al.[60]. Briefly, 15 μl of *P. protegens* CHA0 cell-free filtrate was incubated with 25 μl of azocasein (2% w/v in 50 mM Tris-HCl pH 8.0) at 40 °C for 24 h. One hundred and twenty-five of 10% w/v cold trichloroacetic acid (TCA) was added to precipitate superfluous azocasein, and then 100 μl supernatant was neutralized with 100 μl of 1 M NaOH after centrifugation at 5000*g* for 30 min. Optical density read at 440 nm was used as a proxy of exoprotease activity.

*Tryptophan side chain oxidase (TSO) activity*. TSO activity, an indicator of quorum sensing activity in *P. protegens* CHA0, was measured based on an modified established colorimetric assay[61]: Three-day-old bacterial cultures grown in 1/3 strength liquid KB were mixed at a 1:1 ratio with a reagent solution (5 g l$^{-1}$ SDS, 37.6 g l$^{-1}$ glycine 2.04 l$^{-1}$ g tryptophan, pH 3.0) and TSO activity was measured as optical density at 600 nm after overnight incubation.

*Biofilm formation*. We quantified bacterial biofilm formation using a standard protocol[62]. Briefly, bacteria were grown at 20 °C for 72 h in 160 μl 1 g l$^{-1}$ TSB in a 96-well microtiter plate with TSP lid (TSP, NUNC, Roskilde, Denmark). Planktonic cells were removed by immersing the lid with pegs three times in phosphate-buffered saline solution (PBS). Subsequently, the biofilm on the pegs was stained for 20 min in 160 μl 1% crystal violet solution. Pegs were washed five times in PBS after which the crystal violet was extracted for 20 min from the biofilm in a new 96-well microtiter plate containing 200 μl 96% ethanol per well. Biofilm formation was defined as the optical density at 590 nm of the ethanol extracted crystal violet[63].

*Inhibition of other microorganisms*. Antimicrobial activity was defined as the relative growth of the target organism in *P. protegens* supernatant compared to the control treatment. Antifungal activity of the cell-free supernatant was assessed against the ascomycete *Verticillium dahliae*. The fungus was grown on potato dextrose agar at 28 °C for 4 days, after which plugs of fungal mycelium were incubated in potato dextrose broth medium at 28 °C and gentle shaking for 5 days. Fungal spores were collected by filtering out the mycelium from this culture over glass wool. Subsequently, spores were washed and resuspended in water and the OD595 of the suspension was adjusted to 1. Five microlitres of this spore suspension was then inoculated with 15 μl *P. protegens* CHA0 cell-free filtrate and incubated in 160 μl of 1 g l$^{-1}$ PDB medium for 2 days at 20 °C in 96-well plates. Fungal growth was measured as optical density at 595 nm after 2 days of growth and contrasted with the growth in the control treatment (PDB medium without *P. protegens* supernatant). Antibacterial activity was determined using the plant pathogen *Ralstonia solanacearum* as a target organism. *R. solanacearum* was grown in 160 μl of 1 g l$^{-1}$ TSB medium supplemented with 15 μl of *P. protegens* CHA0 cell-free filtrate or 15 μl of 1/3 strength liquid KB as a control for 2 days at 20 °C. *R. solanacearum* growth was measured as optical density at 600 nm.

**Determining changes in *P. protegens* CHA0 interactions with *A. thaliana* after the selection experiment**. Based on the life-history trait measurements, five distinct bacterial phenotypes were identified using *K*-means clustering analysis (Fig. S2). In order to assess whether phenotypic changes reflected shifts in the strength and type of plant–bacterium interaction, we chose five isolates from each bacterial phenotype group representing each replicate selection line and five ancestral isolates for further measurements (a total of 30 isolates, Table S2).

*Effects of ancestor and evolved bacteria on plant performance*. For each isolate we measured root colonizing ability and impact on plant performance. All 30 bacterial isolates were incubated overnight in 1/3 KB strength liquid at 20 °C. The culture was centrifuged twice for 5 min at 5000 × *g* and the pellet was washed and finally resuspended in 10 mM MgSO4. The resulting suspension was adjusted to an OD600 of 0.01 for each strain[64]. Ten microlitres of the bacterial suspension (or 10 mM MgSO4 as a control) was applied to the roots of three 10-day old sterile *Arabidopsis thaliana* Col-0 seedlings (excluding 2 days of stratification at 4 °C) grown on vertically positioned Petri dishes with agar-solidified (1.5% agar (w/v)) modified Hoagland's medium (*n* = 3 biological plant replicates, each containing 3 seedlings). Plants were grown for 14 days before harvesting. Plants were photographed before and 14 days after bacterial inoculation.

Bacterial effects on plant health were quantified as leaf 'greenness' as the presence of ancestral strain was observed to lead to bleaching and loss of chlorophyll in A. thaliana leaves. The 'greenness' was quantified from photographs by measuring the number of green pixels. To this end, photographs were first transformed in batch

using Adobe Photoshop 2021 by sequentially selecting only green areas followed by thresholding balancing green tissue over background noise (Level 80). This resulted in black-and-white images for further analysis, and the mean number of white pixels per fixed-sized region-of-interest of the aboveground tissue was subsequently determined as 'greenness' using ImageJ (version 1.50i). The numbers of lateral roots and the primary root length were also measured using ImageJ. The root morphology data measured at the end of the experiment was normalized with the data collected at the time of inoculation for each individual seedling.

To determine shoot biomass, the rosette of each plant was separated from the root system with a razor blade and weighted. The roots were placed into a pre-weighted 1.5 ml Eppendorf tubes to quantify the root biomass. To determine the bacterial abundance per plant, these tubes were subsequently filled with 1 ml 10 mM MgSO4 buffer solution and two glass beads. The rhizosphere bacteria were suspended into the buffer solution using a TissueLyser II at a frequency of 20 s$^{-1}$ for 1 min after which bacterial densities were determined using flow cytometry (BD Accuri™ C6 Plus, thresholds for FSC: 2000, SSC: 8000). Shoot biomass, root biomass, root length and number of lateral roots were used in a principal component analysis (PCA) to calculate an overall impact of the bacteria on plant performance (Fig. 2e). The first principal component (PC1) explained 79.9% of the variation and was normalized against the control treatment to be used as a proxy of 'Plant performance' in which positive values reflect plant growth promotion and negative values plant growth inhibition.

*Root derived carbon source utilization*. To measure changes in bacterial growth on potential root derived carbon sources, we measured the growth of all 256 isolates using modified Ornston and Stanier (OS) minimal medium[65] supplemented with single carbon sources at a final concentration of 0.5 g l$^{-1}$ in 96-well plates containing 160 μl selected OS medium per well. The following carbon sources were selected based on their relatively high abundance in *Arabidopsis* root exudates[21]: alanine, arabinose, butyrolactam, fructose, galactose, glucose, glycerol, glycine, lactic acid, putrescine, serine, succinic acid, threonine and valine. Bacterial growth was determined by measuring optical density at 600 nm after 3 days incubation at 20 °C.

*GUS histochemical staining assay and bacterial growth under scopoletin stress*. To investigate effects of the ancestor and evolved strains of *P. protegens* CHA0 on expression of *MYB72*, we applied a *GUS* histochemical staining assay to the 30 selected isolates (Table S2). MYB72 is a transcription factor involved in production of the coumarin scopoletin in *Arabidopsis* roots and specific rhizobacteria can upregulate expression of *MYB72* in the roots[66]. Scopoletin is an iron-mobilizing phenolic compound with selective antimicrobial activity[22]. Seedlings of the *A. thaliana* MYB72$_{pro}$:GFP-GUS[24] reporter line were prepared as described above. Seven-day-old seedlings were inoculated directly below the hypocotyls with 10 μl of a bacterial suspension (OD660 = 0.1)[24]. At 2 days after inoculation, the roots were separated from the shoots and washed in MilliQ water (Milliport Corp., Bedford, MA) to remove all the adhered bacteria. GUS staining of the roots was performed in 12-well microtiter plates where each well contained roots of 5–6 seedlings and 1 ml of freshly prepared GUS substrate solution (50 mM sodium phosphate with a pH at 7, 10 mM EDTA, 0.5 mM K4[Fe(CN)6], 0.5 mM K3[Fe (CN)6], 0.5 mM X-Gluc, and 0.01% Silwet L-77)[67]. Plates were incubated in the dark at room temperature for 16 h. The roots were fixed overnight in 1 ml ethanol: acetic acid (3:1 v/v) solution at 4 °C and transferred to 75% ethanol. Then the pictures of each microtiter plates were taken, and GUS activity was quantified by counting the number of blue pixels in each well of the microtiter plates using image analysis in ImageJ (version 1.52t). To assess the effects of scopoletin on ancestral and evolved *P. protegens* CHA0 isolates, we applied a sensitivity assay to the 30 selected isolates (Table S2). In brief, growth of bacterial isolates was measured in 1 g l$^{-1}$ TSB medium (160 μl) supplemented with scopoletin at final concentrations of 0 μM (control), 500 μM, 1000 μM, and 2 mM using optical density at 600 nm after 72 h incubation at 20 °C without shaking in 96-well microtiter plates. Maximal effect (Emax) of scopoletin was calculated via R package 'GRmetrics'[68] as an indication of scopoletin tolerance.

**Whole-genome sequencing**. All 30 isolated phenotypes were whole genome sequenced to identify possible mutations and affected genes. To this end, isolates were cultured overnight at 28 °C in 1/3 strength liquid KB. Chromosomal DNA was isolated from each culture using the GenElute™ Bacterial Genomic DNA Kit Protocol (NA2100). DNA samples were sheared on a Covaris E-220 Focused-ultrasonicator and sheared DNA was then used to prepare Illumina sequencing libraries with the NEBNext® Ultra™ DNA Library Prep Kit (New England Biolabs. France) and the NEBNext® Multiplex Oligos for Illumina® (96 Index Primers). The final libraries were sequenced in multiplex on the NextSeq 500 platform (2 × 75 bp paired-end) by the Utrecht Sequencing Facility (http://www.useq.nl) yielding between 1.0 and 6.4 million reads per sample equivalent to ~10–70-fold coverage (based on comparison with the original 6.8 Mbp reference genome NCBI GenBank: CP003190.1).

**Variant calling analysis**. We first constructed an updated reference genome of *P. protegens* CHA0, carrying the *GFP* marker gene on its chromosome, from the ancestral strain using the A5 pipeline with default parameters[69]. The input dataset

for this sample consisted of 3,1M reads and totals an approximate 34-fold coverage. The size of the updated reference genome is 6.8 Mbp, with a G + C content of 63.4%, and it comprises 80 scaffolds, with a $N_{50}$ value of 343 kbp. We subsequently used PROKKA[70] (version 1.12; https://github.com/tseemann/prokka) for full annotation of the updated reference genome, and this resulted in the identification of 6147 genes. The updated genome is deposited in NCBI GenBank with following reference: RCSR00000000.1.

Having established the ancestral genome sequence, we subsequently used Snippy (version 3.2-dev; https://github.com/tseemann/snippy) to identify and functionally annotate single-nucleotide polymorphisms and small insertions and deletions (indels) for each individual strain. In addition, we investigated the breadth of coverage for each gene per sample with BedTools[71] to identify genes with large insertions or deletions. An overview of the polymorphisms is shown in Supplementary Table S3. Raw sequencing data for this study are deposited at the NCBI database under BioProject PRJNA473919.

**Relative competitive fitness of *gac* mutants measured in vivo and in vitro**. The relative competitive fitness of selected *gac* mutants was measured in direct competition with their direct ancestors both in vivo in the rhizosphere of *A. thaliana* and in vitro in different standard culture media. Relative fitness was measured as deviation from initial 1:1 ratio of bacterial clone pairs based on PCR-based high-resolution melting profile (RQ-HRM) analysis. Three pairs of isolates were selected: (A) evolved *gacA* ID 242 (genotype *oafA*$^{Y335X}$ · *RS17350*$^{A77A.fsX14}$ · *gacA*$^{D49Y}$) and its direct ancestral genotype 133 (genotype *oafA*$^{Y335X}$ · *RS17350*$^{A77A.fsX14}$) from evolutionary line 1; (B) evolved *gacA* ID 220 (genotype *galE*$^{V32M}$ · *accC*$^{E413K}$ · *gacA*$^{D54Y}$) and its direct ancestral genotype 28 (genotype *galE*$^{V32M}$ · *accC*$^{E413K}$) from line 2; (c) evolved *gacS* ID 222 (genotype *oafA*$^{K338S.fsX18}$ · *gacS*$^{G27D}$) and its direct ancestral genotype 66 (genotype *oafA*$^{K338S.fsX18}$) from line 3. Bacterial isolates were first grown overnight in KB medium at 28 °C, centrifuged at 5000*g* for 10 min and the pellet resuspended in 10 mM $MgSO_4$. This washing procedure was repeated twice. The resulting bacterial suspensions were diluted to $OD_{600} = 0.05$. The initial inoculum for the competition assays was then generated by mixing equal volumes of evolved and ancestral competitors in a ratio of 1:1.

*Measuring competitive fitness in* A. thaliana *rhizosphere*. This assay was performed on the roots of 10-day old *A. thaliana* seedlings grown on full strength Hoagland agar plates, which were prepared as described earlier. Twenty microlitres of the initial inoculum, containing a total of $10^6$ bacterial cells, was inoculated on to the root–shoot junction of each seedling. After 14 days of growth, bacterial populations were isolated from the roots and stored at −80 °C in 42.5% glycerol for relative abundance measurements.

*Measuring competitive fitness in culture media*. Competition assays were also performed in three commonly used nutrient-rich growth media: KB, LB and TSB. KB contained 20 g proteose peptone, 1.5 g $MgSO_4.7H_2O$, 1.2 g $KH_2PO_4$ and 10 g glycerol per litre and the pH was adjusted to 7.3 ± 0.2. TSB contained 30 g tryptic soy broth per litre and pH was adjusted to 7.3 ± 0.2. LB contained 10 g peptone, 5 g yeast extract and 5 g NaCl per litre. Twenty microlitres inoculum of competing strains, containing about $10^6$ bacterial cells, were added into wells containing 140 µl fresh medium in a 96-well plate. The microplates were incubated at 28 °C without shaking for 48 after 80 µl sample was harvested and stored at −80 °C in 42.5% glycerol from each well for relative abundance measurements.

*RQ-HRM assay for quantifying changes in genotype frequencies after competition*. We used a high-resolution melting (HRM) curve profile assay with integrated LunaProbes to quantify the ratio of mutant to wild-type genotypes[72–74]. The probes and primers used in this study are listed in Table S4. Primers were designed using Primer3. Probes were designed with the single-nucleotide polymorphism (SNP) located in the middle of the sequence, and the 3′ end was blocked by carbon spacer C3. The primer asymmetry was set to 2:1 (excess primer: limiting primer) in all cases. Pre-PCR was performed in a 10-µl reaction system, with 0.25 µM excess primer, 0.125 µM limiting primer, 0.25 µM probe, 0.5 µl bacterial sample culture (100-fold diluted saved sample, $OD_{600}$ is about 0.01), 1× LightScanner Master Mix (BioFire Defense). DMSO with the final concentration 5% was supplemented in all reactions to ensure the targeted melting domains are within the detection limit of the LightScanner (Idaho Technology Inc.). Finally, MQ water was used to supplement up to 10 µl. A 96-well black microtiter plate with white wells was used to minimize background fluorescence. Before amplification, 25 µl mineral oil was loaded in each well to prevent evaporation, and the plate was covered with a foil seal to prevent the degradation of fluorescent molecules. Amplification was initiated by a holding at 95 °C for 3 min, followed by 55 cycles of denaturation at 95 °C for 30 s, annealing at 60 °C for 30 s and extension at 72 °C for 30 s and then kept at 72 °C for 10 min. After amplification, samples were heated in a ThermalCycler (Bio-Rad) shortly to 95 °C for 30 s to denature all double-stranded structures followed by a rapid cooling to 25 °C for 30 s to facilitate successful hybridization between probes and the target strands. The plate was then transferred to a LightScanner (Idaho Technology Inc.). Melting profiles of each well were collected by monitoring the continuous loss of fluorescence with a steady increase of the temperature from 35 to 97 °C with a ramp rate of 0.1 °C/s. The relative quantification was based on the negative first derivative plots using software MATLAB.

The areas of probe-target duplexes melting peaks were auto-calculated by 'AutoFit Peaks I Residuals' function in software PeakFit (SeaSolve Software Inc.). The mutant frequency X was calculated using the formula shown below:

$$X = \frac{Area_{mutant}}{Area_{mutant} + Area_{WT}} \quad (1)$$

To validate the RQ-HRM method, standard curves were generated by measuring mixed samples with known proportions of mutant templates: 0, 10, 20, 30, 40, 50, 60, 70, 80, 90 and 100%. Measurements for each sample were done in triplicate. Linear regression formula of each mutant between actual frequencies and measured frequencies are shown in Fig. S7. The high $R^2$ values, and nearly equal to 1 slope values of these equations, confirmed that the RQ-HRM method can accurately detect mutants' frequency in a mixed population.

The relative fitness of the evolved strains was calculated according to previous studies using the following equation[75,76]:

$$relative\ fitness(r) = \frac{X_1(1 - X_0)}{X_0(1 - X_1)} \quad (2)$$

where $X_0$ is the initial mutant frequency; $(1−X_0)$ the initial ancestor frequency; $X_1$ the final mutant frequency; and $(1−X_1)$ is the final ancestor frequency.

**Reporting summary**. Further information on research design is available in the Nature Research Reporting Summary linked to this article.

## Data availability

The *Pseudomonas protegens* CHA0-GFP reference strain genome sequence, determined for this study, is deposited on GenBank with accession RCSR00000000.1. Raw sequencing data used in this study are deposited at the NCBI database under BioProject PRJNA473919. Raw data of *P. protegens* CHA0 phenotypic traits are described in Supplementary Data 1–3, which are also deposited at Mendeley Data: https://doi.org/10.17632/3g6db3pj6b.2. Source data are provided with this paper.

## Code availability

R code is deposited on GitHub (https://github.com/erqinli/plant-rhizosphere-mutualist; https://doi.org/10.5281/zenodo.4750789). Source data are provided with this paper.

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

## Acknowledgements
We thank Roy van der Meijs for his excellent work on the scopoletin sensitivity assays and Ke Yu, Roeland Berendsen and members of the Plant-Microbe Interactions Lab for helpful discussions. This work was supported by a China Scholarship Council fellowship (to E.L.) and a postdoctoral fellowship of the Research Foundation Flanders (FWO 12B8116RN) (to R.d.J.). V.-P.F. is funded by the Royal Society (RSG\R1\180213 and CHL \R1\180031) and jointly by a grant from UKRI, Defra, and the Scottish Government, under the Strategic Priorities Fund Plant Bacterial Diseases programme (BB/T010606/1) at the University of York.

## Author contributions
E.L., P.A.H.M.B., C.M.J.P. and A.J. designed the experiments. E.L., H.J. and C.L. performed the experiment. E.L., R.d.J., V.-P.F. and A.J. analysed the data. All authors collegially wrote the manuscript.

## Competing interests
The authors declare no competing interests.
