## [Peer Review File · Nature Communications]

Reviewers' Comments:

Reviewer #1:

Remarks to the Author:

This manuscript describes experiments in which the plant-associated bacterium *Pseudomonas protegens* CHA0 was grown in association with *Arabidopsis thaliana* under gnotobiotic conditions for six 1-month cycles. The resulting rhizobacterial populations were harvested at two month intervals and analyzed for the "evolution" of a number of phenotypic traits, many of which were related to mutations in the GacS/GacA two component regulatory system. In many respects, the experimental system was not different from growth of the bacteria in a Petri dish and indeed, it is well-known that Gac mutants do arise with significant frequency upon growth in vitro on rich media. What is different is that the authors posit that the plant exerts selection on the bacteria for mutualists, and that in at least one specific case, there is selection for enhanced tolerance to scopoletin, produced by the plant. Despite the extensive phenotypic characterization carried out by the authors, who offer credible possibilities to explain their results, I think there are too many alternative explanations for the observed results to make the manuscript acceptable in its current form. For example, in lines 264-266, the authors speculate that the reduced production of exometabolites by the bacteria had a positive effect on the availability of plant-derived nutrients. I would suggest that the energy saved by the bacteria not producing the exometabolites might simply have made them able to grow with greater energy efficiency. With regard to the upregulation of *myb72*, the authors did not demonstrate that more scopoletin was synthesized as a function of increased expression of the regulatory gene, nor (if I interpret it correctly) did Fig 3C show increased tolerance to scopoletin. In lines 176-178, it was not clear to me that the same isolates were increasing the expression of *myb72* and exhibiting increased tolerance to that metabolite.

In conclusion, I do agree the manuscript will provoke thinking in the field. The authors have demonstrated that it is becoming feasible to consider undertaking such such onerous and technically demanding experiments, but I also think that more details must be addressed to make the conclusions convincing. And if the experiments were to be reproduced, perhaps it might make sense to work with a strain like *P. simiae*, which makes fewer exometabolites than CHA0. It is noteworthy that the synthesis of exometabolites controlled by Gac can affect *myb* expression.

Reviewer #2:

Remarks to the Author:

In their manuscript "Rapid evolution of bacterial mutualism in the plant rhizosphere", the authors experimentally evolved isolates of *Pseudomonas protegens* that initially were plant pathogens, conferring net costs to the host *Arabidopsis thaliana*. Yet, after six plant growth cycles, mutualist isolates of *P. protegens* evolved that significantly increased plant growth compared to uninoculated control plants. The authors tracked the evolution of novel variants over the six growth cycles for five independent plant lines, and used K-means clustering analysis to identify five distinct phenotypes that emerged. They also completed whole genome sequencing on a subset of the isolates and found novel mutations within the GacS/GacA pathway, suggesting that the shift from parasitism to mutualism only required a small number of mutations in global regulator genes.

Overall, the authors conducted a solid study that shows a really exciting and novel result. To my knowledge, the authors are the first to show that an initially pathogenic strain of *Pseudomonas* evolves to be beneficial when passaged with a plant host. Because *Pseudomonas* is a common soil microbe, I believe their results to be general, in that they likely play-out in natural systems with other plant hosts. I also felt the authors' approach to identify several distinct phenotypes based on many different traits (growth, production of compounds, antimicrobial activity) was novel and made their results both compelling and easy to follow. I especially loved Fig. 2A that captures when the different phenotypes emerged in the passaging experiment and how the frequencies of those phenotypes changed across five independent passaging lines. My feedback detailed below is more about the

wording of their main interpretation, expanding on the discussion, and minor clarifications of some of the methodologies, rather than the soundness of the results reported.

General feedback

In the intro, the authors set up the hypothesis that plant discrimination mechanisms could drive the evolution of mutualistic microbes, and argue that their results support this hypothesis (e.g., Lines 225-226: "Together, these results show that plant selection can lead to high level of parallel evolution both at the phenotypic and molecular level."; also see lines 247-249, 268-271, and 283-285). My concern is that they cannot rule out the alternative (and perhaps, more interesting) hypothesis that mutualism was driven by microbial adaptation to the rhizosphere, which happens to benefit the plant. The importance for distinguishing between these hypotheses is how much we expect coevolution to play out in this system. Because the authors only used one genotype of the plant, and plants in their experiment did not evolve, their results are consistent with microbial adaptation to plant roots, and thus, do not require that plants have evolved discrimination mechanisms to select for more beneficial microbes. Instead of being plant-driven, the phenotypic and genetic changes the authors observed were microbially-driven via adaptation to the rhizosphere; those that adapted were competitively superior in the rhizosphere compared to free-living conditions, as Fig. 5 clearly shows. The most interesting aspect of their study to me was that the mutations underlying the switch to mutualism were found in genes underlying global regulation, suggesting that pleiotropy (one gene affecting many traits) could be extensive, and thus, might provide the link between competitive superiority and plant-growth promotion mechanisms. For example, is there any evidence that fold induction of MYB72 on its own increases plant growth? I.e., if one were to induce expression of this gene in the absence of microbes, would plants grow larger compared to those that do not express it? Although further experiments would have to be conducted, it would be super interesting to me if this were true (i.e., induction of MYB72 and plant growth were positively correlated), because this would provide a strong link between microbial competitive superiority (via a trait like Scopoletin tolerance), and plant growth promotion via induction of MYB72. To summarize, I recommend a simple re-wording throughout to emphasize that microbial adaptation to the rhizosphere is what drives the transition to mutualism, and then expand on a discussion about how such adaptation happens to benefit the plant via pleiotropy for both competitive superiority in the rhizosphere and plant growth-promotion mechanisms.

Minor comments

Line 79: five independent *A. thaliana* Col-0 replicate plant selection lines. Does "line" here mean individual replicate, or genotype? It would be helpful to clarify whether the different lines used were replicated of the same genotype Col-0, given that "line" is used interchangeably with "genotype" in the literature.

Line 101: "measured all" - replace with "measuring several"

Line 170: add "s" after plant

Line 184: subset of evolved isolates - state how many here, even if it's in the methods/figures.

Line 260: add "s" after effect

Whole Genome Sequencing (starting at line 592): it would be useful to know whether *Pseudomonas protegens* is known to harbor any mobile genetic elements such as plasmids or chromosomal islands, and relatedly, whether presence/absence variation in genes existed among the isolates sequenced. This would allow the reader to determine that the novel mutations that arose (i.e., presented in Fig. 4) did so within a common genetic background, rather than being linked to a gene that was only present in some isolates but not others.

Figure 1F: outlier points are larger than those in A-E, should make consistent

Figure 4

legend (line 368): add "of" after out

Clarify here and in the methods whether all novel mutations appear in the figure, or only a subset. It would also be helpful to get a sense of allele-frequency for each mutation (i.e., the proportion of isolates that possess that particular mutation), which could be added to the table.

Overall, I greatly enjoyed reading their well-written and compelling manuscript, and once published, I believe others who are interested in microbiome research and the evolution of mutualism will find it to be an important contribution as well.

Sincerely,

Rebecca Batstone

Reviewer #3:

Remarks to the Author:

This paper describes a nice and convincing work showing that antagonistic bacteria can evolve into mutualistic ones (beneficial effect on plant root and shoot biomass) when they are serially submitted to conditions under which they need the plant to survive (plant exudates as sole carbon compounds). This transition is accompanied by improved bacterial fitness in the rhizosphere. The paper is clear and well written.

That the bacteria increase their fitness in conditions under which they were evolved is expected. What is less expected –and this makes the originality of the work- is that the bacteria also enhance the plant fitness along the experiment. There is thus an alignment between bacterial and plant fitness, although bacteria are not plant endophytes (clearly shown in fig2 panel C).

Two aspects seem important to understand, i) how evolved bacteria positively affects plant growth and ii) how plant and bacteria fitness are linked.

i). A first way is the decrease of bacterial production of cytotoxic compounds, which is the case of mutualistic 2 clones. But how mutualistic 1 clones affect plant growth? Authors should provide hypotheses at least in the discussion.

ii). It is proposed that the increases in plant growth leads to the production of more plant exudates and thus favours the growth of bacteria. It is more efficient if this trait is coupled with better use of plant compounds otherwise it would led to the proliferation of a bacterial cheating population that only better use plant exudates (case of most M1 clones). But how mutualists 2 (which do not better use plant compounds) benefit from better plant growth? This could be discussed.

To better evidence the links between plant performance and bacterial traits, it would have been relevant to provide figures showing the correlation between plant performance and each of these traits (proteolytic activity, C use, scopoletin tolerance) with the clones colored in function of their group (such as fig3 panel D). Indeed, the paper figures generally show the phenotypic traits in function of bacterial groups (transient, mutualistic1 etc..), but bacteria are grouped in function of life-history traits ((growth, stress tolerance...unrelated to plant performance) and mutualists groups contain clones that have no impact on plant growth (ID151) or impact the plant similarly to clones of other groups (ID242). Since two mutualist clones that have a strong plant effect have a single mutation (gac-40T>A in M1 ID188 and gacY183S in M2 ID172), it would be interesting to indicate these two clones on the correlation figures, and thus highlight the role of these gac mutations in all studied traits (plant performance and bacterial traits) and in coupling plant and bacterial fitness.

The role of the gac regulatory genes (as described in the literature) could be more discussed in regards with the different phenotypes observed.

Minor comments

P3 I50. "detrimental". I am not aware of rhizobia that decrease legume growth compared to non-inoculated control.

P10|205-206. « while 'Mutualist 2' isolates showed a severe to complete disruption of secondary metabolite production (Fig. S3) ». Mutualists 2 showed a strong decrease in proteolytic activity, not exactly in II metabolite production.

P10, I213. " mutualists evolved in all except one selection line, which became dominated by 'Stress-sensitive' bacteria (Fig. 1, Fig. 4)". Mutualists were found in all lines see fig2A and Table S2, but they only invade 4 out of the 5 lines, likely because they emerged lately.

P12 I257 "potentially" or demonstrated? See ref 20.

P12 I259. "leading to clear phenotypic and genetic bacterial diversification" sounds strange for me since the change in the plant-bacteria interaction is driven by genetic diversification and selection. I would have written " ...this interaction rapidly evolved during the experiment as bacterial mutualists that had positive effect on plant growth and relatively higher competitive advantage in the rhizosphere compared to ancestors emerged and invaded the bacterial population".

P12 line 26 "reduction in the production of exoproducts, including lytic enzymes and antimicrobial...". This doesn't seem to be the case for mutualists 1.

P12 "In turn, improved plant growth likely triggered selection for mutualists that were better at competing for root exudates relative to other phenotypes, or by selectively constraining the growth of non-mutualist phenotypes via certain sanctioning mechanisms". Bacterial culture in plant rhizosphere triggers selection for bacteria that are better at competing for root exudates, not necessarily mutualists. I don't think the logic for "in turn improved plant growth". "selectively constraining the growth of non-mutualist phenotypes via certain sanctioning mechanisms" is strange. Which sanctions? Please clarify.

P12 I275 "predicted functional effects of observed gac mutations " . Could you explain?

P12 I275 "together these adaptations could have created a strong selective advantage for mutualistic phenotypes". Bacteria with better C use or better scopoletin tolerance but with no or negative impact on plant growth could have emerged and proliferated.

This paragraph could be improved by providing clearer explanations. For me there are several possible strategies for root-associated bacteria to improve their fitness, including: i) increase their growth competitiveness in the rhizosphere via better use of exudates or better scopoletin tolerance or better biofilm formation, ii) enhance plant growth and as result benefit from more carbon sources, or iii) both enhance scopoletin production by the plant and scopoletin tolerance (all these phenotypes are found among evolved bacteria). Evolved bacteria that invade the root-associated population after 6 cycles are bacteria that combine plant growth promotion and one or several other strategies, likely because it is more efficient. This was obtained via mutations in the gacS/gacA regulatory genes. Could the authors point out mutations that both improve plant growth and competitiveness (either C use or scopoletin tolerance)?

Authors also should provide hypothesis on how mutualists 1 -that do not decrease cytotoxic compound production- impact plant growth.

P13 I278 "with only a single or a few successive" instead of "with only a few successive"

P13 I286-287. It could also be due to a late emergence of a gacA/gacS beneficial mutation.

P17 line 336 "around" rather than "in" the plant

P18. Panel C fig3. I don't understand the y axis. Scopoletin tolerance represents DO scopoletin2000/DO scopoletin 0. In dataset2 sheet 3 DO scopoletin2000 is higher than DO scopoletin 0 for (at least) the first ID (7,16,21). I can't see this on the figure.

P18 fig3. To show that improved bacterial fitness is due to increased capacity to grow on plant exudates and/or increased capacity to tolerate scopoletin, it would have been relevant to show the relationship between bacterial abundance (cells per plant) and growth on root secreted C sources or increased capacity to tolerate scopoletin. It would have been interesting to indicate on all the correlation figures the two clones that bear a unique mutation in or upstream gacA (mutualist1 ID188 and mutualist 2 ID172, which both have the strongest effects on plant performance), thus evidencing the link between these gacA mutations and the different phenotypic traits investigated. This genotype-phenotype correlation should be emphasized in paragraph "Mutualistic phenotypes had mutations in genes encoding the GacA/GacS regulatory system" since it demonstrates the adaptive role of the gacA mutations.

It would be better to differently color ancestral and ancestral-like clones (this will allow to distinguish them in correlation figures).

P20, fig4. There are two figures. They should be named panel A and panel B and commented as such in the legend.

Panel A:

"Filled dots represent isolates with non-synonymous mutations (present in 18/25 evolved isolates)": Only 18 out of the 20 evolved clones represented in panel A have non-synonymous mutations, the two others (mutualistic 1 and 2 clones) have mutations upstream a gene (gacA-40T>A).

There is an inversion between the transient clone oafAY335X RS17350 and the ancestral-like clone oafAY335X RS17350 wbpM clone.

x-axis: Scale is different between fig4 and figS5.

Panel B

"The table lists unique mutations (and the strains' ID number) linked with evolved bacterial phenotypes, and additional mutations that appeared later during the experiment within the same genetic background are shown after the indent; notably, these additional mutations did not affect the bacterial phenotypes ». ID number is missing in the table. In line 5 the stress-sensitive clone with mutations rpoS and tetR appeared to be more performant than the clone with the sole rpoS mutation suggesting that the TetR mutation increases the effect on plant performance.

Mutualistic 1 line2:"gacAg97S" instead of "gacAY97S".

RS17350A77A.fsX14?

P46. « resistance » instead of « resistace » in the y-axis of panels B and C(figS4).

P47; figS5. The relative abundance of evolved phenotypes is not explained in Figure 1. How is measured the frequency of each isolate in their end population?

In correlation graphs, it is difficult to distinguish the dark grey of ancestral and the darkgreen of mutualistic 2. Could you change the colors?

P51. Table S2. Some mutations (Gac-40T>A, mraZ-211A>G, hult786C>T) are not nonsynonymous mutations but mutations upstream a gene.

I don't understand the nomenclature oafAK338.fsX18 (which might correspond to c.1009A deleted/early stop in table S3) and RS17350A77A.fsX14 (which should correspond with c.116C deleted/early stop in Table S3).

X represents a stop codon, while in table S3 * represents a stop codon. Please homogenize.

REVIEWER COMMENTS

Reviewer #1 (Remarks to the Author):

This manuscript describes experiments in which the plant-associated bacterium *Pseudomonas protegens* CHA0 was grown in association with *Arabidopsis thaliana* under gnotobiotic conditions for six 1-month cycles. The resulting rhizobacterial populations were harvested at two month intervals and analyzed for the “evolution” of a number of phenotypic traits, many of which were related to mutations in the GacS/GacA two component regulatory system. In many respects, the experimental system was not different from growth of the bacteria in a Petri dish and indeed, it is well-known that Gac mutants do arise with significant frequency upon growth in vitro on rich media. What is different is that the authors posit that the plant exerts selection on the bacteria for mutualists, and that in at least one specific case, there is selection for enhanced tolerance to scopoletin, produced by the plant. Despite the extensive phenotypic characterization carried out by the authors, who offer credible possibilities to explain their results, I think there are too many alternative explanations for the observed results to make the manuscript acceptable in its current form.

1. Response: We thank reviewer #1 for constructive comments. As suggested also by reviewer #2, we have now refined the interpretation of our results and included some additional analyses showing positive relationship between plant performance and MYB72 induction (Figure 3E). Briefly, we now conclude that while bacterial adaptation in the rhizosphere likely drove the increase in its genetic diversity, both bacterial competitive differences, and active plant discrimination via MYB72-mediated scopoletin production, could have driven observed changes in *P. protegens* CHA0 genotype frequencies over time.

For example, in lines 264-266, the authors speculate that the reduced production of exometabolites by the bacteria had a positive effect on the availability of plant-derived nutrients. I would suggest that the energy saved by the bacteria not producing the exometabolites might simply have made them able to grow with greater energy efficiency. s

2. Response: This is a good additional hypothesis, which we now discuss on lines 288-295. We also now include additional data where we show that bacterial abundances were negatively correlated with ‘antagonistic traits’ associated with antibiosis with plant and fungi (Figure S7), suggesting that reduced antagonism was correlated with enhanced bacterial growth.

With regard to the upregulation of *myb72*, the authors did not demonstrate that more scopoletin was synthesized as a function of increased expression of the regulatory gene, nor (if I interpret it correctly) did Fig 3C show increased tolerance to scopoletin. In lines 176-178, it was not clear to me that the same isolates were increasing the expression of *myb72* and exhibiting increased tolerance to that metabolite.

3. Response: We have previously established a strong relationship between scopoletin production under increased MYB72 induction in this system (Stringlis *et al.* 2018), and hence, scopoletin production by plants was not quantified in these experiments. We have now clarified the relationship between scopoletin production and the other traits and show that there is a positive relationship between MYB72 activation and scopoletin tolerance (Figure 3D) and that plant performance was positively correlated with MYB72 induction (Figure 3E), and negatively correlated with proteolytic activity (Figure 3F).

In conclusion, I do agree the manuscript will provoke thinking in the field. The authors have demonstrated that it is becoming feasible to consider undertaking such onerous and technically demanding experiments, but I also think that more details must be addressed to make the conclusions convincing. And if the experiments were to be reproduced, perhaps it might make sense to work with a strain like *P. simiae*, which makes fewer exometabolites than CHA0. It is noteworthy that the synthesis of exometabolites controlled by Gac can affect myb expression.

4. Response: We thank reviewer #1 for kind words. We agree that *P. simiae* would be a great candidate strain for further research to study if similar evolutionary responses are observed with bacterial taxa that produce fewer secondary metabolites. We also now briefly discuss the link between Gac-mediated MYB72 expression in the discussion along with new additional results shown in Figure 3 (on lines 329-336).

Reviewer #2 (Remarks to the Author):

In their manuscript “Rapid evolution of bacterial mutualism in the plant rhizosphere”, the authors experimentally evolved isolates of *Pseudomonas protegens* that initially were plant pathogens, conferring net costs to the host *Arabidopsis thaliana*. Yet, after six plant growth cycles, mutualist isolates of *P. protegens* evolved that significantly increased plant growth compared to uninoculated control plants. The authors tracked the evolution of novel variants over the six growth cycles for five independent plant lines, and used K-means clustering analysis to identify five distinct phenotypes that emerged. They also completed whole genome sequencing on a subset of the isolates and found novel mutations within the GacS/GacA pathway, suggesting that the shift from parasitism to mutualism only required a small number of mutations in global regulator genes.

Overall, the authors conducted a solid study that shows a really exciting and novel result. To my knowledge, the authors are the first to show that an initially pathogenic strain of *Pseudomonas* evolves to be beneficial when passaged with a plant host. Because *Pseudomonas* is a common soil microbe, I believe their results to be general, in that they likely play-out in natural systems with other plant hosts. I also felt the authors' approach to identify several distinct phenotypes based on many different traits (growth, production of compounds, antimicrobial activity) was novel and made their results both compelling and easy to follow. I especially loved Fig. 2A that captures when the different phenotypes emerged in the passaging experiment and how the frequencies of those phenotypes changed across five independent passaging lines. My feedback detailed below is more about the wording of their main interpretation, expanding on the discussion, and minor clarifications of some of the methodologies, rather than the soundness of the results reported.

5. Response: We thank reviewer #2 for kind words and very helpful comments.

General feedback

In the intro, the authors set up the hypothesis that plant discrimination mechanisms could drive the evolution of mutualistic microbes, and argue that their results support this hypothesis (e.g., Lines 225-226: “Together, these results show that plant selection can lead to high level of parallel evolution both at the phenotypic and molecular level.”; also see lines 247-249, 268-271, and 283-285). My concern is that they cannot rule out the alternative (and perhaps, more interesting) hypothesis that mutualism was

driven by microbial adaptation to the rhizosphere, which happens to benefit the plant. The importance for distinguishing between these hypotheses is how much we expect coevolution to play out in this system. Because the authors only used one genotype of the plant, and plants in their experiment did not evolve, their results are consistent with microbial adaptation to plant roots, and thus, do not require that plants have evolved discrimination mechanisms to select for more beneficial microbes. Instead of being plant-driven, the phenotypic and genetic changes the authors observed were microbially-driven via adaptation to the rhizosphere; those that adapted were competitively superior in the rhizosphere compared to free-living conditions, as Fig. 5 clearly shows.

6. Response: We fully agree with the reviewer and have now clarified that our results were solely driven by bacterial evolution as plants were not let evolve in our experiment. Our revised reasoning is following.

We conclude that the evolutionary outcome was driven by bacterial diversification in the plant rhizosphere environment, which provided different ecological opportunities (niches) and favoured evolved bacterial genotypes over the ancestral strain. In four out of five plant selection lines, this led to increase in the frequency of bacterial genotypes that showed reduced antagonism to the plant ('Mutualists'). These 'Mutualists' showed improved ability to grow on plant root exudates and had increased tolerance to plant-derived scopoletin antimicrobial, which was positively linked with increased MYB72 induction that governs scopoletin production. MYB72 induction had also direct positive effects on plant growth. Crucially, the fitness benefits of mutualists were not observed in lab media under mere resource competition and likely explanation for this is that scopoletin tolerance increased mutualist fitness mainly in the presence of plant.

Together, these results suggest that mutualists evolved in response to two ecological opportunities in the plant rhizosphere: a) 'root exudate niche' and b) 'antibiotic tolerance niche'. Both adaptations were linked with reduced antagonistic activity and increased MYB72 induction, which was directly beneficial for the plant growth, creating a positive feedback loop for the bacterial growth.

The most interesting aspect of their study to me was that the mutations underlying the switch to mutualism were found in genes underlying global regulation, suggesting that pleiotropy (one gene affecting many traits) could be extensive, and thus, might provide the link between competitive superiority and plant-growth promotion mechanisms. For example, is there any evidence that fold induction of MYB72 on its own increases plant growth? I.e., if one were to induce expression of this gene in the absence of microbes, would plants grow larger compared to those that do not express it?

7. Response: The gene MYB72 plays an integral part of the plants adaptive strategy to iron deficiency and survival in alkaline soils where iron availability is largely restricted (Stringlis *et al.* 2018). Moreover, MYB72 can protect plants from the infection by several pathogens via activation of induced systematic response (Pieterse *et al.* 2020). As a result, mutants that trigger increase in MYB72 expression could also have positive selection on the plant growth. In support for this hypothesis, we identified a significant positive relationship between plant performance and MYB72 induction ($R^2=0.5$, $P < 0.001$; Figure 3E), which suggest that evolved mutualists showed a direct positive effect on plant growth in addition to reduction in antagonism (now discussed on lines 295-312).

Although further experiments would have to be conducted, it would be super interesting to me if this were true (i.e., induction of MYB72 and plant growth were positively correlated), because this would

provide a strong link between microbial competitive superiority (via a trait like Scopoletin tolerance), and plant growth promotion via induction of MYB72.

8. Response: Please, see our response above regarding additional analysis. We now also report negative correlations between evolved clones' plant performance and 'antagonistic traits' (proteolytic and antifungal activity: Figure 3F, Figure S6E-F) to highlight the pleiotropic effects underlying *P. protegens* adaptation. We further propose in the discussion that GacS/GacA could act as an important gene determining whether *P. proteges* CHAO engages in plant growth-promotion, virulence or competitive microbial interactions.

To summarize, I recommend a simple re-wording throughout to emphasize that microbial adaptation to the rhizosphere is what drives the transition to mutualism, and then expand on a discussion about how such adaptation happens to benefit the plant via pleiotropy for both competitive superiority in the rhizosphere and plant growth-promotion mechanisms.

9. Response: We have now modified the manuscript as suggested (see also our response 6).

Minor comments

Line 79: five independent *A. thaliana* Col-0 replicate plant selection lines. Does "line" here mean individual replicate, or genotype? It would be helpful to clarify whether the different lines used were replicated of the same genotype Col-0, given that "line" is used interchangeably with "genotype" in the literature.

10. Response: With 'line' we refer to independent plant replicates (5 in total; all Col-0). We have now specified that each plant replicate line was initiated using the same ancestral Col-0 genotype (on lines 77-80).

Line 101: "measured all" - replace with "measuring several"

11. Response: Revised accordingly.

Line 170: add "s" after plant

12. Response: Revised accordingly.

Line 184: subset of evolved isolates - state how many here, even if it's in the methods/figures.

13. Response: Revised accordingly.

Line 260: add "s" after effect

14. Response: Revised accordingly.

Whole Genome Sequencing (starting at line 592): it would be useful to know whether *Pseudomonas protegens* is known to harbor any mobile genetic elements such as plasmids or chromosomal islands, and relatedly, whether presence/absence variation in genes existed among the isolates sequenced. This would allow the reader to determine that the novel mutations that arose (i.e., presented in Fig. 4) did so within a common genetic background, rather than being linked to a gene that was only present in some isolates but not others.

15. Response: We have now included more detail about the mobile genetic elements present in *P. protegens* CHA0 genome. Furthermore, we have specified in the methods and results that no movement of mobile genetic elements were observed in our evolved clones.

Figure 1F: outlier points are larger than those in A-E, should make consistent

16. Response: Revised accordingly.

Figure 4 legend (line 368): add “of” after out

17. Response: Revised accordingly.

Clarify here and in the methods whether all novel mutations appear in the figure, or only a subset. It would also be helpful to get a sense of allele-frequency for each mutation (i.e., the proportion of isolates that possess that particular mutation), which could be added to the table.

18. Response: The frequency of key mutations is described in the main text and individual mutants are described in more detail in the Table S3. As Figure 4 focuses on presenting the genetic basis and parallel evolution of mutualist and stress-sensitive phenotypes, it works less well for presenting allele frequencies for every observed mutation. Moreover, as we only sequenced a subset of clones instead of conducting population level sequencing, our data is better at describing the link between specific mutations and phenotypes instead of reliably presenting allele frequencies within selection lines.

Overall, I greatly enjoyed reading their well-written and compelling manuscript, and once published, I believe others who are interested in microbiome research and the evolution of mutualism will find it to be an important contribution as well.

Sincerely,
Rebecca Batstone

19. Response: We thank again reviewer #2 for kind words and very helpful comments.

Reviewer #3 (Remarks to the Author):

This paper describes a nice and convincing work showing that antagonistic bacteria can evolve into mutualistic ones (beneficial effect on plant root and shoot biomass) when they are serially submitted to conditions under which they need the plant to survive (plant exudates as sole carbon compounds). This transition is accompanied by improved bacterial fitness in the rhizosphere. The paper is clear and well written.

That the bacteria increase their fitness in conditions under which they were evolved is expected. What is less expected –and this makes the originality of the work- is that the bacteria also enhance the plant fitness along the experiment. There is thus an alignment between bacterial and plant fitness, although bacteria are not plant endophytes (clearly shown in fig2 panel C).

Two aspects seem important to understand, i) how evolved bacteria positively affects plant growth and ii) how plant and bacteria fitness are linked.

i). A first way is the decrease of bacterial production of cytotoxic compounds, which is the case of mutualistic 2 clones. But how mutualistic 1 clones affect plant growth? Authors should provide hypotheses at least in the discussion.

20. Response: We have now toned down the dichotomy between mutualist 1 and 2 genotype differences as considerable variation exists within both groups. For example, in case of proteolytic activity, also some mutualist 1 genotypes showed reduced activity, while this change was more consistent among mutualist 2 genotypes.

As suggested by other reviewers, we have now included two new panels in Figure 3 (E and F). In these figures, we show that plant performance is positively correlated with MYB72 induction (Figure 3E) and negatively correlated with proteolytic activity (Figure 3F). Together, these results show that in addition to reduced production of cytotoxic compounds, mutualists had also direct beneficial effect on the plant growth via MYB72 induction.

ii). It is proposed that the increases in plant growth leads to the production of more plant exudates and thus favours the growth of bacteria. It is more efficient if this trait is coupled with better use of plant compounds otherwise it would led to the proliferation of a bacterial cheating population that only better use plant exudates (case of most M1 clones). But how mutualists 2 (which do not better use plant compounds) benefit from better plant growth? This could be discussed.

21. Response: We fully agree that mutualist bacteria should be favoured by the plant – otherwise bacterial populations would be susceptible to the invasion by cheaters, which would reduce the frequency of mutualistic genotypes. We have now clarified this process with both mutualist 1 and 2 genotypes.

In case of mutualist 1, we now show that improved ability to grow on plant exudates was coupled with MYB72 induction that had positive effect on plant growth (Figure 3E). This could create a positive discrimination for mutualist 1 genotype over ancestral strain (as shown by competition assays in Figure 5) and constrain the emergence of cheaters as they would have negative effect on plant growth and root exudation.

In case of mutualist 2, we show that improved ability to tolerate scopoletin was coupled with MYB72 induction, which had positive effects on plant growth and scopoletin production (Figure 3D-E). As a result, MYB72 induction mediated by scopoletin tolerant mutualist 2 genotypes also likely led to a positive discrimination over ancestral strain (as shown by competition assays in Figure 5).

Both mutualist adaptations were coupled with reduced plant antagonism, which further strengthened positive feedback with plants and mutualists.

We also propose in the discussion that mutualists 1 and 2 could have reinforced their coexistence in replicates 2 and 4 via MYB72 induction, that likely positively discriminated them both over other genotypes by providing a 'nutrient niche' for mutualist 1, and a 'scopoletin tolerance niche' for mutualist 2 genotypes.

To better evidence the links between plant performance and bacterial traits, it would have been relevant to provide figures showing the correlation between plant performance and each of these traits

(proteolytic activity, C use, scopoletin tolerance) with the clones colored in function of their group (such as fig3 panel D). Indeed, the paper figures generally show the phenotypic traits in function of bacterial groups (transient, mutualistic1 etc.), but bacteria are grouped in function of life-history traits ((growth, stress tolerance...

22. Response: We have now included two new panels to Figure 3 (E and F) to show correlations between MYB72 induction with plant performance and proteolytic activity. Furthermore, other life-history trait correlations are presented in two new supplementary figures (Figure S6 and S7).

unrelated to plant performance) and mutualists groups contain clones that have no impact on plant growth (ID151) or impact the plant similarly to clones of other groups (ID242). Since two mutualist clones that have a strong plant effect have a single mutation (*gac-40T>A* in M1 ID188 and *gacY183S* in M2 ID172), it would be interesting to indicate these two clones on the correlation figures, and thus highlight the role of these *gac* mutations in all studied traits (plant performance and bacterial traits) and in coupling plant and bacterial fitness.

23. Response: We have added more text on lines 222-231, to describe these different mutations' effect on plant growth. As shown in the Figure 4, ID 251 (*fleQ* (R320Q)) is not promoting plant growth, and only GacS (G27D) and GacA (D49Y) showed a lower level of plant growth promotion (all other *gac* mutants showed positive effects on plant). We have now highlighted four clones (ID172, ID188, ID242 and ID 251) in correlation figures and discuss their characteristics in the results section.

The role of the *gac* regulatory genes (as described in the literature) could be more discussed in regards with the different phenotypes observed.

24. Response: We have now added a new section to the discussion where we briefly discuss how Gac regulatory genes might be linked to metabolism, reduced antagonism, MYB72 induction and scopoletin tolerance (lines 295-312).

Minor comments

P3 I50. "detrimental". I am not aware of rhizobia that decrease legume growth compared to non-inoculated control.

25. Response: Certain rhizobial strains can be considered as parasitic if they fix little or no nitrogen for the plant but still reap benefits of receiving carbon. For example, see review by Denison and Kiers 2004.

P10I205-206. « while 'Mutualist 2' isolates showed a severe to complete disruption of secondary metabolite production (Fig. S3) ». Mutualists 2 showed a strong decrease in proteolytic activity, not exactly in II metabolite production.

26. Response: We have toned down this section and refer these activities as extracellular instead of mediated by secondary metabolism. The sentence has been revised as follows: '*while 'Mutualist 2' isolates showed a severe to complete disruption of extracellular proteolytic and antifungal activity (Figure S3)*'.

P10, I213. "mutualists evolved in all except one selection line, which became dominated by 'Stress-sensitive' bacteria (Fig. 1, Fig. 4)". Mutualists were found in all lines see fig2A and Table S2, but they only invade 4 out of the 5 lines, likely because they emerged lately.

27. Response: We have revised this section and make distinction that while mutualists evolved in all selection lines, they became dominant in only 4 out of 5 replicates. We have included the late emergence of mutualists as one potential explanation for this result.

P12 I257 “potentially” or demonstrated? See ref 20.

28. Response: We have removed the word ‘potentially’ and cite reference 20 instead.

P12 I259.”leading to clear phenotypic and genetic bacterial diversification” sounds strange for me since the change in the plant-bacteria interaction is driven by genetic diversification and selection. I would have written “ ...this interaction rapidly evolved during the experiment as bacterial mutualists that had positive effect on plant growth and relatively higher competitive advantage in the rhizosphere compared to ancestors emerged and invaded the bacterial population”.

29. Response: We have now revised the text as suggested.

P12 line 266 “reduction in the production of exoproducts, including lytic enzymes and antimicrobial...”. This doesn’t seem to be the case for mutualist 1.

30. Response: We have now clarified that while mutualist 1 genotypes showed a reduction in their proteolytic activity, this was not as clear as with mutualist 2 genotypes.

P12 “In turn, improved plant growth likely triggered selection for mutualists that were better at competing for root exudates relative to other phenotypes, or by selectively constraining the growth of non-mutualist phenotypes via certain sanctioning mechanisms”. Bacterial culture in plant rhizosphere triggers selection for bacteria that are better at competing for root exudates, not necessarily mutualists. I don’t think the logic for “in turn improved plant growth”. “selectively constraining the growth of non-mutualist phenotypes via certain sanctioning mechanisms” is strange. Which sanctions? Please clarify.

31. Response: We have now clarified in the text that MYB72 induction was also linked to scopoletin tolerance. As MYB72 induction has previously been shown to be linked to increased scopoletin production by the plant, it could have ‘sanctioned’ the non-tolerant genotypes, while positively discriminating mutualist 2 genotypes.

P12 I277 “predicted functional effects of observed gac mutations “ . Could you explain?

32. Response: We have included citations to make it more explicit that we refer to previous work on functional effects of Gac mutations by others (references 28-31)

P12 I278 “together these adaptations could have created a strong selective advantage for mutualistic phenotypes”. Bacteria with better C use or better scopoletin tolerance but with no or negative impact on plant growth could have emerged and proliferated. This paragraph could be improved by providing clearer explanations. For me there are several possible strategies for root-associated bacteria to improve their fitness, including: i) increase their growth competitiveness in the rhizosphere via better use of exudates or better scopoletin tolerance or better biofilm formation, ii) enhance plant growth and as result benefit from more carbon sources, or iii) both enhance scopoletin production by the plant and scopoletin tolerance (all these phenotypes are found among evolved bacteria). Evolved bacteria that invade the root-associated population after 6 cycles are bacteria that combine plant growth promotion and one or several other strategies, likely because it is more efficient. This was obtained via mutations in

the *gacS/gacA* regulatory genes. Could the authors point out mutations that both improve plant growth and competitiveness (either C use or scopoletin tolerance)?

33. Response: We fully agree and have now clarified this in the manuscript. Briefly, with new analyses, we show that traits that likely increased bacterial competitiveness (growth on exudates and scopoletin tolerance) were positively linked with plant growth via MYB72 induction and reduced antagonism (mutualist strategy; please see also our response to comment 6). However, we also pinpoint that this was not always the case as seen with stress-sensitive genotypes that evolved very efficient biofilm formers via alternative evolutionary trajectory.

Authors also should provide hypothesis on how mutualists 1 –that do not decrease cytotoxic compound production- impact plant growth.

34. Response: We now conclude that mutualist 1 genotypes had a positive effect on plant growth via MYB72 induction and reduction in cytotoxic activity (Figure 3E-F). While these effects were not as extreme as with mutualist 2 genotypes, they were significantly different from the ancestral strain.

P13 I278 “with only a single or a few successive” instead of “with only a few successive”

35. Response: Revised accordingly.

P13 I286-287. It could also be due to a late emergence of a *gacA/gacS* beneficial mutation.

36. Response: We have now included this explanation to the discussion.

P17 line 336 “around” rather than “in” the plant

37. Response: Revised accordingly.

P18. Panel C fig3. I don't understand the y axis. Scopoletin tolerance represents DO scopoletin2000/DO scopoletin 0. In dataset2 sheet 3 DO scopoletin2000 is higher than DO scopoletin 0 for (at least) the first ID (7,16,21). I can't see this on the figure.

38. Response: The Y-axis shows change in bacterial growth in the presence of scopoletin at 72h time point relative to the initial background OD. OD600 values on the first 361 rows show start point values (just after inoculation, 'time'=0). These values are about 0.035, showing the background OD of the culture media and plastic bottom of 96-well plates. The OD values reflecting bacterial growth after 72h (starting from row 722, 'time'=72) reach about 0.3 in the absence of scopoletin and about 0.2 in the presence of scopoletin. We have now described axes more clearly in the Figure 3.

P18 fig3. To show that improved bacterial fitness is due to increased capacity to grow on plant exudates and/or increased capacity to tolerate scopoletin, it would have been relevant to show the relationship between bacterial abundance (cells per plant) and growth on root secreted C sources or increased capacity to tolerate scopoletin. It would have been interesting to indicate on all the correlation figures the two clones that bear a unique mutation in or upstream *gacA* (mutualist1 ID188 and mutualist 2 ID172, which both have the strongest effects on plant performance), thus evidencing the link between these *gacA* mutations and the different phenotypic traits investigated. This genotype-phenotype correlation should be emphasized in paragraph “Mutualistic phenotypes had mutations in genes encoding the GacA/GacS regulatory system” since it demonstrate the adaptive role of the *gacA*

mutations. It would be better to differently color ancestral and ancestral-like clones (this will allow to distinguish them in correlation figures).

39. Response: We have now included correlations between scopoletin tolerance and growth on root exudates with plant performance in supplementary materials (Figures S6 and S7). We have also indicated the effect of the four unique mutants in these graphs as suggested.

P20, fig4. There are two figures. They should be named panel A and panel B and commented as such in the legend. Panel A: “Filled dots represent isolates with non-synonymous mutations (present in 18/25 evolved isolates)”: Only 18 out of the 20 evolved clones represented in panel A have non-synonymous mutations, the two others (mutualistic 1 and 2 clones) have mutations upstream a gene (*gacA*-40T>A).

40. Response: Revised accordingly.

There is an inversion between the transient clone *oafAY335X* RS17350 and the ancestral-like clone *oafAY335X* RS17350 *wbpM* clone.

41. Response: We have fixed this information in the figure.

x-axis: Scale is different between fig4 and fig5.

42. Response: This is because the X-axis in Fig5 shows the frequency of different genotypes at the end of the selection experiment (an indicator of each evolved subpopulation’s fitness during the selection experiment) and is not derived from the plant performance fitness assays as in Figure 4.

Panel B: “The table lists unique mutations (and the strains’ ID number) linked with evolved bacterial phenotypes, and additional mutations that appeared later during the experiment within the same genetic background are shown after the indent; notably, these additional mutations did not affect the bacterial phenotypes ». ID number is missing in the table. In line 5 the stress-sensitive clone with mutations *rpoS* and *tetR* appeared to be more performant than the clone with the sole *rpoS* mutation suggesting that the TetR mutation increases the effect on plant performance.

43. Response: We have now revised this table and removed the strains ID numbers. We agree that the additional *TetR* mutations seems to have epistatic effect with *rpoS* mutation – however, this effect was not statistically significant. We now mention this in the results section.

Mutualistic 1 line2: “*gacAg97S*” instead of “*gacAY97S*”.

44. Response: Revised accordingly.

RS17350A77A.fsX14?

45. Response: We have provided more information to explain the way we present the mutation effects on lines 1078-1084: ‘For example, *flhA*^{H393Q.fsX15} means this mutation (a single deletion) lead to an amino acid change to Q from H at position 393 due to the frame shift (fs) caused by the deletion followed by a stop codon X after another 15 amino acids (X15). Notably, these additional amino acids are in a different frame and thus represent a totally different sequence than the wild type allele.’

P46. « resistance » instead of « resistace » in the y-axis of panels B and C(figS4).

46. Response: Revised accordingly.

P47; figS5. The relative abundance of evolved phenotypes is not explained in Figure 1. How is measured the frequency of each isolate in their end population?

47. Response: We apologise for a typo; we meant the figure 2A. The relative frequency of each phenotype was determined by determining the ration of individuals in each phenotypic group compared to the total number of cultured isolates/cycle/line (16).

In correlation graphs, it is difficult to distinguish the dark grey of ancestral and the dark green of mutualistic 2. Could you change the colors?

48. Response: We have now updated the colours in all figures using black for 'Ancestor', dark grey for 'Ancestral-like', grey for 'Transient' and orange for 'Stress-sensitive'. Mutualists 1 and 2 are still presented as light and dark green colours. Please note, these colour changes should also make figures more accessible for colour blind readers.

P51. Table S2. Some mutations (Gac-40T>A, mraZ-211A>G, hult786C>T) are not non-synonymous mutations but mutations upstream a gene.

49. Response: Yes, Gac-40T>A and mraZ-211A>G are mutations upstream a gene, and this is why we are using A, T, C and G to present the changes. The hult786C>T, however, is a synonymous mutation, with C mutated to T at sequence position 786 without changing the amino acid. These details are presented in Table S3 in the 'Effect' column.

I don't understand the nomenclature oafAK338.fsX18 (which might correspond to c.1009A deleted/early stop in table S3) and RS17350A77A.fsX14 (which should correspond with c.116C deleted/early stop in Table S3). X represents a stop codon, while in table S3 * represents a stop codon. Please homogenize.

50. Response: Yes, this interpretation is correct. We have now updated the Table S3 using consist style of presentation throughout. Also, we have added some more information to explain the way we present the effects of mutations (on lines 1087-1088); please see our response above.

References:

Stringlis, I. A. *et al.* MYB72-dependent coumarin exudation shapes root microbiome assembly to promote plant health. *Proc. Natl. Acad. Sci. U. S. A.* **115**, E5213–E5222 (2018).

R. Ford Denison, E. Toby Kiers, Lifestyle alternatives for rhizobia: mutualism, parasitism, and forgoing symbiosis, *FEMS Microbiology Letters*, Volume 237, Issue 2, August 2004, Pages 187–193,

Reviewers' Comments:

Reviewer #1:

Remarks to the Author:

This manuscript describes experimental evolution, over a period of 6 months in a gnotobiotic system, of *Pseudomonas protegens* grown on *Arabidopsis*. The bacteria demonstrated increased mutualistic fitness via improved competitiveness for root exudates and enhanced tolerance to the plant antimicrobial compound scopoletin, and the increased fitness traits were expressed only in planta. The results showed that the bacteria were capable of rapid evolution along a parasitism-mutualism continuum.

The manuscript is a toned-down version of the original submission, with better explanation of the data, more figures, much less speculation in the discussion of the results, and it was a pleasure to read. I found a typo on line 180 (where should be were) but little else to complain about.

Reviewer #2:

Remarks to the Author:

In their revised manuscript, the authors refined their interpretations by honing in on two non-mutually exclusive explanations: differences in bacterial competitive fitness and MYB72-mediated plant discrimination. They added a very helpful result (Fig 3E) showing that MYB72 induction was positively correlated with plant performance, suggesting that such induction is pleiotropically associated with both bacterial competitive advantage and plant benefit. The authors clarify that their results did not require plant adaptation/evolution; rather root responses already present in their plant lines were enough to drive the shift from parasitism to mutualism in the bacteria populations. With these clarifications and added results to support their hypotheses, I believe their manuscript is in great shape for publication.